# Single-molecule turnover dynamics of actin and membrane coat proteins in clathrin-mediated endocytosis

Michael M Lacy[1,2,3], David Baddeley[2,4†], Julien Berro[1,2,4]*

[1]Department of Molecular Biophysics and Biochemistry, Yale University, New Haven, United States; [2]Nanobiology Institute, Yale University, West Haven, United States; [3]Integrated Graduate Program in Physical and Engineering Biology, Yale University, New Haven, United States; [4]Department of Cell Biology, Yale University School of Medicine, New Haven, United States

**Abstract** Actin dynamics generate forces to deform the membrane and overcome the cell's high turgor pressure during clathrin-mediated endocytosis (CME) in yeast, but precise molecular details are still unresolved. Our previous models predicted that actin filaments of the endocytic meshwork continually polymerize and disassemble, turning over multiple times during an endocytic event, similar to other actin systems. We applied single-molecule speckle tracking in live fission yeast to directly measure molecular turnover within CME sites for the first time. In contrast with the overall ~20 s lifetimes of actin and actin-associated proteins in endocytic patches, we detected single-molecule residence times around 1 to 2 s, and similarly high turnover rates of membrane-associated proteins in CME. Furthermore, we find heterogeneous behaviors in many proteins' motions. These results indicate that endocytic proteins turn over up to five times during the formation of an endocytic vesicle, and suggest revising quantitative models of force production.

*For correspondence:
julien.berro@yale.edu

Present address: †Auckland Bioengineering Institute, University of Auckland, Auckland, New Zealand

Competing interests: The authors declare that no competing interests exist.

## Introduction

Clathrin-mediated endocytosis (CME) is the major pathway for eukaryotic cells to internalize plasma membrane and extracellular molecules. In yeast, the invagination of a clathrin-coated pit (CCP) and formation of the ~50 nm vesicle relies on a dense meshwork of cytoskeletal actin filaments and associated proteins (*Goode et al., 2015*; *Kaksonen et al., 2003*; *Kaksonen et al., 2006*). This localized, rapidly-assembled actin meshwork is necessary to generate the force to bend the plasma membrane inward and overcome the cell's high turgor pressure (*Aghamohammadzadeh and Ayscough, 2009*; *Boulant et al., 2011*; *Collins et al., 2011*). Although many endocytic proteins and their biochemical mechanisms have been characterized, our understanding of the molecular machinery is incomplete, especially in quantitative terms of how the endocytic assembly generates sufficient forces (*Lacy et al., 2018*).

CME is a challenging target for study because the entire assembly is smaller than the optical diffraction limit (~250 nm), the numbers of molecules at the site rapidly change, and most of the dynamics occur within ~20 s. These unique challenges have made theoretical modeling and novel imaging techniques necessary for detailed study of CME (*Berro and Lacy, 2018*). Quantitative microscopy studies of the assembly and disassembly of endocytic proteins have revealed a robust timeline of self-assembly, as actin and associated proteins polymerize into a meshwork during invagination of the CCP and then gradually disassemble after the vesicle is pinched off (*Berro and Pollard, 2014a*; *Kaksonen et al., 2003*; *Picco et al., 2015*; *Sirotkin et al., 2010*; *Taylor et al., 2011*). And while recent super-resolution imaging and electron microscopy of CME sites have revealed increasingly detailed structural organization (*Arasada et al., 2018*; *Mund et al., 2018*; *Sochacki et al.,*

*2017*), these techniques are not currently able to quantify the dynamics of individual molecules within the endocytic site.

Theoretical calculations estimate that the amount of force needed to counteract turgor pressure and invaginate the membrane during CME in yeast is in the range of 1000–3000 pN, or a total energy of $6*10^4$ $k_B$T (*Carlsson and Bayly, 2014*; *Dmitrieff and Nédélec, 2015*; *Lacy et al., 2018*; *Tweten et al., 2017*; *Wang and Carlsson, 2017*; *Ma and Berro, 2019a*). The endocytic actin meshwork is composed of 100–200 short, Arp2/3-branched actin filaments that are capped shortly after nucleation; quantitative modeling indicates fewer than ten filament ends are polymerizing at any given time (*Berro and Pollard, 2014a*; *Sirotkin et al., 2010*), each capable of producing around 1–10 pN of force (*Footer et al., 2007*; *Kovar and Pollard, 2004*). Even after accounting for other force-producing mechanisms, the force from a burst of actin polymerization appears to be insufficient to drive membrane deformation.

Based on the dendritic nucleation model and theoretical results (*Berro et al., 2010*), we hypothesized that continuous polymerization and disassembly of actin filaments underlies the observed dynamics in CME, which could allow the meshwork to generate more force than a single burst of filament polymerization. Turnover dynamics have been observed in other actin assemblies like lamellipodia (*Pollard and Borisy, 2003*; *Watanabe and Mitchison, 2002*; *Yamashiro et al., 2014*). Fluorescence Recovery After Photobleaching (FRAP) experiments with endocytic actin (*Kaksonen et al., 2003*; *Kaksonen et al., 2005*; *Picco et al., 2015*) and membrane-coat proteins (*Skruzny et al., 2012*; *Wu et al., 2001*) suggest the endocytic machinery is more dynamic than has been appreciated, and previous modeling based on mass action kinetics predicted that the number of molecules for actin nucleators Arp2/3 and WASp consumed over an endocytic event was several times greater than the peak number of molecules present (*Berro et al., 2010*). However, these experimental and theoretical results can be difficult to interpret and cannot be used to directly determine specific rates and residence times. Recent literature still generally assumes the endocytic meshwork assembles and disassembles in discrete phases and current models take a simplified view of actin turnover, expecting a single phase of assembly followed by disassembly after vesicle formation (*Carlsson, 2018*; *Hassinger et al., 2017*; *Mund et al., 2018*; *Picco et al., 2018*; *Wang and Carlsson, 2017*). As has been shown in other actin systems, continuous turnover of filaments allows a network to sustain high amounts of force production over time (*Craig et al., 2012*; *Mak et al., 2016*; *McFadden et al., 2017*; *Raz-Ben Aroush et al., 2017*), converting chemical energy from ATP hydrolysis of actin polymerization into mechanical work over the meshwork's lifetime, among other mechanisms. Recent theoretical work based on the dendritic nucleation model, which relies on actin disassembly and turnover, suggests that the endocytic actin meshwork can sustain forces around 2500 pN, sufficient to achieve the membrane invagination (*Nickaeen et al., 2019*). Turnover of network components like crosslinkers would also enable remodeling of the meshwork, allowing the redistribution of stored elastic energy (*Ma and Berro, 2018*; *Picco et al., 2018*; *Ma and Berro, 2019b*) and higher order visco-elastic mechanisms. However, exchange and turnover of single molecules has not been directly observed in CME because conventional microscopy tools lack the resolution to measure such behavior within a transient and diffraction-limited structure like the endocytic patch.

In this study, we applied a variation of single-molecule speckle microscopy to track individual molecules in CME sites in live fission yeast. For eight target proteins, including actin-meshwork and membrane-coat proteins, we sparsely labeled the proteins in the cell and applied single-molecule localization and tracking analyses. These results provide direct evidence for rapid and continuous turnover of the endocytic meshwork on the timescale of 1–2 s, and their motions reveal heterogeneous behaviors at the molecular level. This observation verifies a key behavior of how the endocytic actin meshwork is able to generate and sustain forces to remodel the membrane in CME.

## Results

### Different hypothetical mechanisms can generate similar bulk measurements

A common interpretation of previous endocytic patch intensity tracking data is that the actin monomers and associated proteins polymerize into a meshwork during the CCP invagination and then

gradually disassemble after the vesicle is pinched off. We hypothesized that the apparent time-course of endocytic actin could result from a dynamic structure with continuous turnover where net growth and net disassembly are achieved by altering the balance between nucleation/polymerization and disassembly rates.

We first performed basic logical simulations to illustrate that such fundamentally different models as all-in-one assembly or continuous turnover could each generate apparent dynamics that resemble previously observed microscopy data (*Figure 1*). In the simplest case, components are recruited and

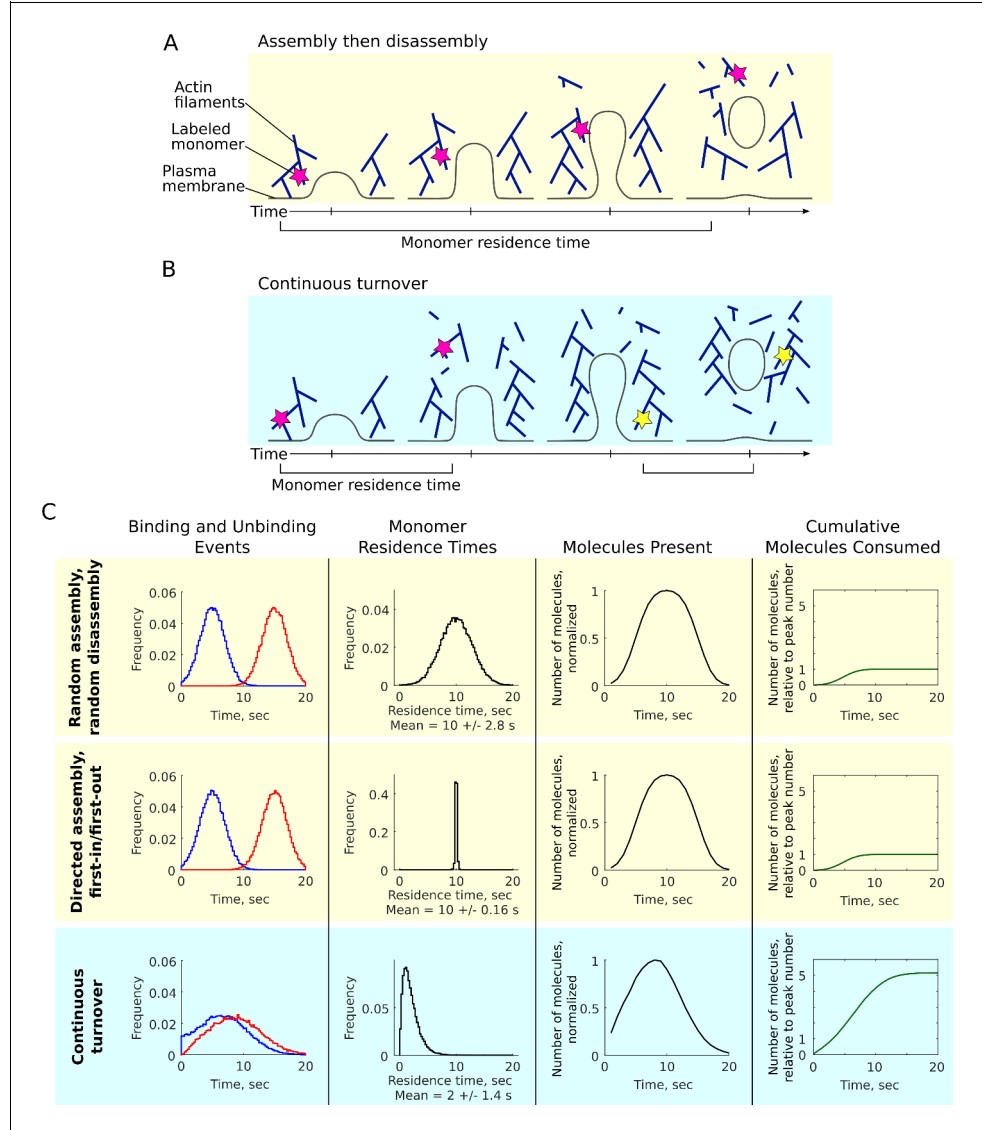

**Figure 1.** Different molecular mechanisms may give rise to similar apparent bulk dynamics. (A–B) Many models suggest directed assembly and disassembly of actin in endocytosis but it is unclear whether the actin meshwork assembles and then disassembles in distinct phases (A) or continuously turns over during its bulk lifetime (B). Individual monomers are illustrated by magenta and yellow stars, with their residence times indicated. (C) Simulated assembly and disassembly according to different hypothetical models: random assembly and disassembly in separate phases (top row), directed assembly with first-in/first-out disassembly (middle row), assembly with continuous turnover (bottom row). Far left: simulated occurrences of binding (blue) and unbinding events (red) underlying the assembly and disassembly mechanisms. Mid left: residence times of individual molecules, a quantity that clearly differentiates between the models. Mid right: number of molecules over time, a measurable quantity that can appear similar across the models. Far right: cumulative number of molecules consumed over time, normalized to the peak number of molecules present.

then disassembled in discrete phases (*Figure 1A*), resulting in the characteristic profile of growth and decay of the number of molecules at the site (*Figure 1C*, top row). A single additional rule, to assume that the oldest components will be the first to be removed ('first-in/first-out', *Figure 1C*, middle row), generates an identical profile of the number of molecules over time. Such a model resembles actin filaments, with polymerization at their barbed end and depolymerization from the pointed end, or other oligomeric lattices with directional assembly and disassembly. A third model of assembly with continuous turnover of short-lived components (*Figure 1B*) can generate a profile of molecules over time that appears similar to the distinct assembly/disassembly models (*Figure 1C*, bottom row).

Although they are not explicit simulations of molecular binding and unbinding processes, these simple logical models provide some illustrative comparisons. The underlying differences in the models are illustrated by the arrival and departure times, or binding and unbinding events (*Figure 1C*, left), features that are not directly accessible by conventional experimental measurements. However, single-molecule observations might reveal the characteristic differences in the residence times of individual objects according to each model (*Figure 1C*, mid right). With random assembly and disassembly the residence times fall in a broad distribution with an average of half the total lifetime (10 s), while for directed assembly with first-in/first-out disassembly, all residence times are equal to half the lifetime (10 s, with some noise). In contrast, continuous turnover results in a distribution of short residence times (2 s) with a peak and tail shape dictated by a combination of underlying kinetic rates.

Importantly, in the discrete assembly models the peak number of molecules contains all of the molecules that participated, while the continuous turnover model consumes a total number of molecules several times greater than the peak number (*Figure 1C*, right). To distinguish between these possible types of behavior in CME, we sought to apply a new single-molecule imaging strategy to directly measure the distributions of residence times in CME.

## Fluorescence recovery after photobleaching indicates rapid recovery of many proteins in CME

We first imaged cells expressing mEGFP-tagged heterodimeric actin capping protein subunit Acp1p in partial-TIRF and tracked endocytic patches, measuring lifetimes of 18.4 ± 3.5 s (mean ± S.D., *Figure 2A–C*, *Table 1*, and *Figure 2—video 1*), consistent with previous reports of capping protein lifetimes (*Berro and Pollard, 2014a*; *Sirotkin et al., 2010*). We also imaged with a scanning confocal microscope and performed Fluorescence Recovery after Photobleaching (FRAP) experiments on four representative endocytic proteins tagged with mEGFP: the Arp2/3 activators Myo1p and Wsp1p, and the actin binding proteins Acp1p and Fim1p (actin filament crosslinker). For all four proteins, the recovery of fluorescence after photobleaching was very fast (*Figure 2D* and *Figure 2—figure supplement 1*, and *Figure 2—video 2*, *Figure 2—video 3*, *Figure 2—video 4*, *Figure 2—video 5*), demonstrating that these proteins exchange rapidly during the assembly and the disassembly stages of endocytic structures. Because the total number of molecules continuously changes during the course of each CME event and we cannot synchronize the precise stage at which bleaching occurs, it is virtually impossible to accurately determine the half-time of fluorescence recovery and the mobile fraction of fluorescence recovered. However, we estimate from our data that the recovery half-times are on the order of 1 s, as the intensity typically reaches the pre-bleached value within 1 to 2 s. The fraction recovered is very large (close to 100%) for all four proteins, and the fluorescence intensity reaches a peak value similar to the peak in unbleached traces (*Figure 2—figure supplement 1*).

## Sparse labeling enables single-molecule imaging in endocytic sites

To make detailed observations of turnover and molecular residence times in the dense assemblies of CME in live fission yeast cells, we employed a strategy based on single-molecule speckle microscopy

**Table 1.** Statistics for tracking of Acp1p-mEGFP endocytic patches.

| Name | Median, sec | Mean, sec | S.D., sec[*] | 95%ile, sec | N tracks | N samples |
|---|---|---|---|---|---|---|
| Acp1-mEGFP | 18.6 | 18.4 | 3.5 | 22.7 | 51 | 2 |

*Standard Deviation is calculated for the distribution of track lifetimes.

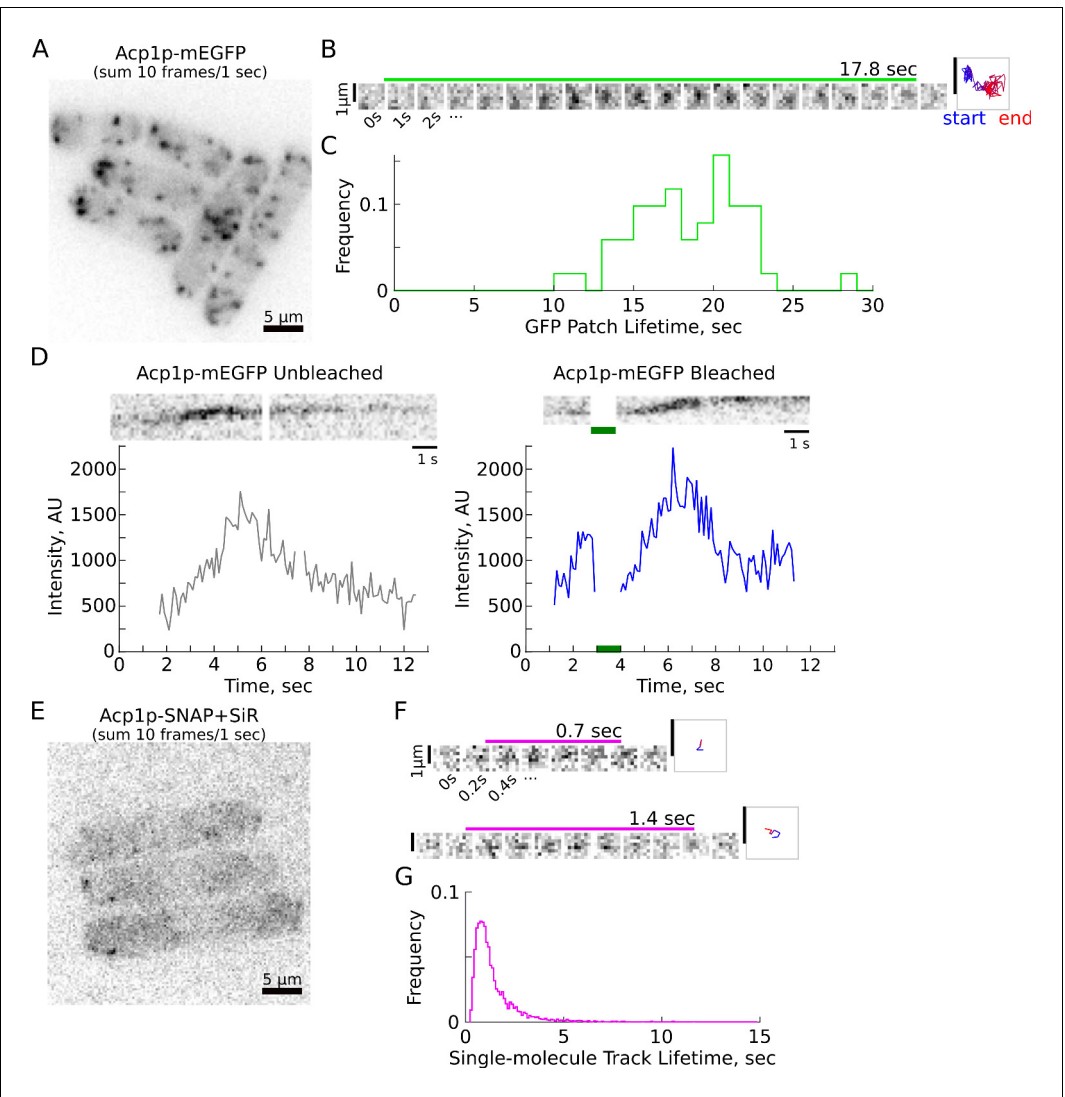

**Figure 2.** Endocytic protein lifetimes assessed by patch tracking, FRAP, and single-molecule tracking. (**A**) Sum intensity projection image of 10 frames (1 s) from a movie of cells expressing Acp1p-mEGFP (heterodimeric actin capping protein subunit) imaged in partial-TIRF (*Figure 2—video 1*), inverted contrast. Scale bar is 5 μm. (**B**) Montage of GFP spot, shown at 1 s increments, with the lifetime of the spot indicated by the green bar above the image panels. Right: trajectory of a spot, color-coded with blue at the start and red at the end. Scale bars are 1 μm. (**C**) Distribution of Acp1p-mEGFP patch lifetimes, as measured by semi-automatic tracking with TrackMate. N = 51 tracks from five movies recorded in two independent samples. (**D**) Acp1p-mEGFP cells were imaged by scanning confocal microscope to record fluorescence recovery after photobleaching of localized regions (*Figure 2—video 5*). Kymographs (above) and background-subtracted fluorescence intensity over time (below) for Acp1p-mEGFP patches (gap in kymograph and intensity spans three or ten frames of localized photobleaching; the unbleached patch is outside of the bleached region but no images are recorded during bleaching). Left: unbleached patch. Right: photobleached patch rapidly recovers to pre-bleached intensity (~1 s) and continues to develop to peak intensity comparable to unbleached patch. Green bar indicates photobleaching pulse. (**E**) Sum projection image of 10 frames (1 s) from a movie of cells expressing Acp1p-SNAP and sparsely labeled with SNAP-SiR imaged in partial-TIRF, inverted contrast (*Figure 2—video 6*). Scale bar is 5 μm. (**F**) Montage of SNAP+SiR spots, shown at 0.2 s increments. Trajectories shown at right (as in C), with scale bars 1 μm. (**G**) Distribution of Acp1p-SiR spot lifetimes, as measured by single-molecule localization and tracking with PYME. N = 4977 tracks from 24 movies recorded in five independent samples. Panels A-B and E-F have been prepared with different brightness settings for visual clarity. See *Figure 2—video 1* and *Figure 2—video 5* for example raw data. The online version of this article includes the following video and figure supplement(s) for figure 2:

**Figure supplement 1.** Fluorescence recovery after photobleaching of endocytic proteins.

*Figure 2 continued on next page*

*Figure 2 continued*

**Figure supplement 2.** Characterization of SNAP-tag labeling.

**Figure 2—video 1.** Acp1-mEGFP The movie was recorded at 10 frames per second and is shown as inverted contrast.

https://elifesciences.org/articles/52355#fig2video1

**Figure 2—video 2.** FRAP of mEGFP-Wsp1p The movie was recorded at 10 frames per second and is shown as inverted contrast.

https://elifesciences.org/articles/52355#fig2video2

**Figure 2—video 3.** FRAP of mEGFP-Myo1p The movie was recorded at 10 frames per second and is shown as inverted contrast.

https://elifesciences.org/articles/52355#fig2video3

**Figure 2—video 4.** FRAP of Fim1p-mEGFP The movie was recorded at 10 frames per second and is shown as inverted contrast.

https://elifesciences.org/articles/52355#fig2video4

**Figure 2—video 5.** FRAP of Acp1p-mEGFP The movie was recorded at 10 frames per second and is shown as inverted contrast.

https://elifesciences.org/articles/52355#fig2video5

**Figure 2—video 6.** Acp1-SNAP+SiR The movie was recorded at 10 frames per second and is shown as inverted contrast.

https://elifesciences.org/articles/52355#fig2video6

**Figure 2—video 7.** Wild type (FY527) + SiR The movie was recorded at 10 frames per second and is shown as inverted contrast.

https://elifesciences.org/articles/52355#fig2video7

---

(*Watanabe and Mitchison, 2002*; *Yamashiro et al., 2014*), an approach that we previously used to reveal dynamics of the yeast eisosome protein Pil1p (*Lacy et al., 2017*). We incubated cells expressing Acp1p fused with the self-labeling SNAP-tag in media with low concentrations of a silicon rhodamine-647-conjugated SNAP substrate (SNAP-SiR) and imaged the cells in partial-TIRF (*Figure 2E* and *Figure 2—video 6*). As reported in *Lacy et al. (2017)*, incubating live fission yeast with low concentrations of SNAP-substrate dye achieves only a low labeling efficiency (see Materials and methods), enabling single-molecule tracking because typically one or zero of these sparsely labeled molecules are incorporated into diffraction-limited CME sites.

We confirmed that the spots observed after SNAP-SiR labeling colocalize specifically with endocytic sites by imaging a strain co-expressing Acp2p-mEGFP and the endocytic actin crosslinker Fim1p-SNAP labeled with SNAP-SiR, and we observed negligible non-specific binding of SNAP-SiR dye in wild-type cells expressing no SNAP-tag (*Figure 2—figure supplement 2*, *Figure 3—video 10*). Our recordings of Acp1p-mEGFP patches (*Figure 2A–C*) demonstrate that we are able to detect the full lifetime of both developing CCPs and vesicles diffusing after membrane scission in the partial-TIRF illumination field. In contrast to the many endocytic patches visible in Acp1p-mEGFP cells (70 to 120 endocytic events occurring per cell at any time *Berro and Pollard, 2014a*), we observe only a few spots in cells expressing Acp1p-SNAP sparsely labeled with SNAP-SiR (referred to as Acp1p-SiR), typically 5 to 10 labeled molecules per cell per minute (*Figure 3—figure supplement 3E*).

## Single-molecule residence times of endocytic proteins are short

We applied super-resolution localization and single-particle tracking (*Baddeley et al., 2011*) to the spots visible in SNAP-SiR labeled samples. Strikingly, Acp1p-SiR spots had short lifetimes in a peaked distribution with a long tail, with average 1.4 s and 95% of events under 3.6 s (*Figure 2F–G*). We imaged nine SNAP-tag strains and generated thousands of tracks for each target protein (*Figure 3* and *Table 2*, *Figure 2—video 6* and *Figure 3—videos 1*, *2*, *3*, *4*, *5*, *6*, *7*, *8*). The eight endocytic proteins included actin (Act1p), actin capping protein subunit (Acp1p), the actin crosslinker fimbrin (Fim1p), an Arp2/3 complex component (Arc5p), myosin-I (Myo1p), the nucleation-promoting factor Wiskott-Aldrich Syndrome Protein (WASp, Wsp1p), clathrin light chain (Clc1p), and the Hip1R/Sla2 homologue and actin-membrane anchoring protein (End4p). We also imaged a non-endocytic protein, the eisosome protein Pil1p, as a control protein that is expected to be immobile

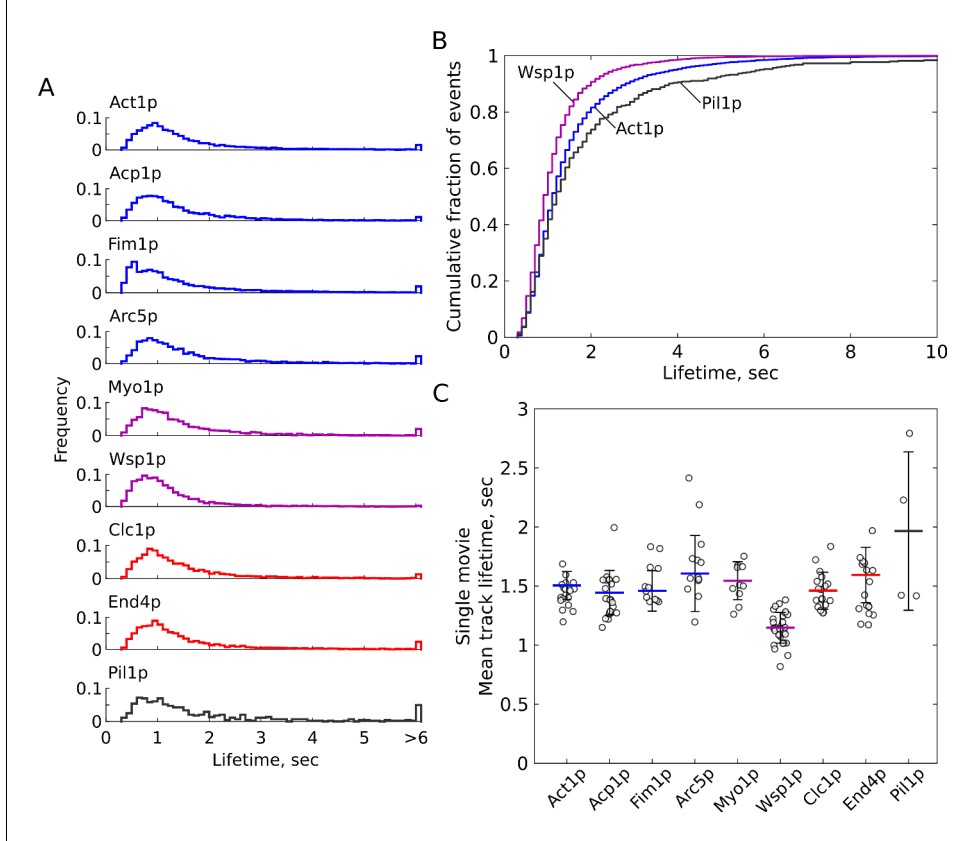

**Figure 3.** Single-molecule residence times of several endocytic proteins. Cells expressing SNAP-tag fusion proteins were sparsely labeled with SiR-647 then imaged and tracked as described in text. (**A**) Probability distributions of track lifetimes for each target protein. Distributions are truncated at 6 s and all events longer than 6 s are shown in the final bin. The minimum allowed track length is 0.3 s. (**B**) Cumulative distributions of Act1p (blue), Wsp1p (purple) and Pil1p (gray). (**C**) Mean track lifetimes calculated for each movie and standard deviation (calculated across means of all images with more than 40 tracks), individual dataset means shown by open circles. See *Table 2* for summary statistics. Actin-associated endocytic proteins are colored in blue, nucleation-promoting factors are purple, and membrane-associated endocytic proteins are red. Pil1p is included as an 'immobile' control in gray, although the detected tracks represent a mixture of stable molecules incorporated in eisosomes and dynamic molecules at eisosome ends. *Figure 2—videos 6, 7, Figure 3—videos 1, 2, 3, 4, 5, 6, 7* are representative movies from which the data shown in this figure have been extracted.

The online version of this article includes the following video and figure supplement(s) for figure 3:

**Figure supplement 1.** Photobleaching does not affect single-molecule track lifetimes.
**Figure supplement 2.** Characterization of single-molecule localization and tracking data.
**Figure supplement 3.** Effects of varying tracking gap-linking parameter.
**Figure supplement 4.** Dependence of track lifetimes on illumination start time cutoff.
**Figure 3—video 1.** Act1p-SNAP+SiR The movie was recorded at 10 frames per second and is shown as inverted contrast.
https://elifesciences.org/articles/52355#fig3video1
**Figure 3—video 2.** Fim1p-SNAP+SiR The movie was recorded at 10 frames per second and is shown as inverted contrast.
https://elifesciences.org/articles/52355#fig3video2
**Figure 3—video 3.** Arc5p-SNAP+SiR The movie was recorded at 10 frames per second and is shown as inverted contrast.
https://elifesciences.org/articles/52355#fig3video3
**Figure 3—video 4.** SNAP-Myo1p+SiR The movie was recorded at 10 frames per second and is shown as inverted contrast.
https://elifesciences.org/articles/52355#fig3video4

**Table 2.** Statistics for single-molecule tracks of SNAP-tag fusion proteins.

| Name | Lifetime Median, sec | Lifetime Mean, sec | Lifetime S.D., sec[*] | Lifetime 95%ile, sec | End-to-end distance Median, nm | n tracks | N movies (datasets with > 40 tracks) | N samples | D, nm²/sec [†] |
|------|------|------|------|------|------|------|------|------|------|
| Act1p | 1.1 | 1.5 | 0.12 | 4 | 159 | 7660 | 18 (18) | 2 | 2955 |
| Acp1p | 1.1 | 1.4 | 0.19 | 3.6 | 167 | 4977 | 24 (21) | 5 | 3874 |
| Fim1p | 1 | 1.5 | 0.17 | 4.2 | 154 | 16,490 | 12 (12) | 2 | 2855 |
| Arc5p | 1.2 | 1.6 | 0.32 | 4.2 | 166 | 4576 | 17 (13) | 2 | 3337 |
| Myo1p | 1.1 | 1.6 | 0.16 | 3.9 | 156 | 3951 | 10 (10) | 2 | 1678 |
| Wsp1p | 0.9 | 1.2 | 0.13 | 2.6 | 147 | 4379 | 41 (32) | 5 | 1869 |
| Clc1p | 1.1 | 1.5 | 0.16 | 3.7 | 168 | 2718 | 19 (18) | 3 | 2770 |
| End4p | 1.1 | 1.6 | 0.24 | 4.1 | 148 | 3983 | 31 (17) | 5 | 1381 |
| Pil1p | 1.2 | 2.0 | 0.67 | 5.9 | 134 | 446 | 6 (4) | 1 | 517 |

[*]Standard Deviation is determined across means calculated from individual datasets (each movie), only counting movies with >40 tracks.

[†]Apparent diffusion coefficient calculated from mean squared-displacement over time for time windows < 1 s (10 points), as in **Figure 5**.

(*Kabeche et al., 2011*; *Walther et al., 2006*). Although a subset of Pil1p-SiR molecules undergoes rapid binding and unbinding at the ends of the eisosome structure (*Lacy et al., 2017*), a much greater fraction of Pil1p-SiR spots persists longer than 5 s and the average lifetime is longer than any of the endocytic proteins (*Figure 3* and *Table 2*).

GFP-tagged versions of actin-associated proteins (Act1p, Acp1p, Fim1p, and Arc5p) have been previously reported to have bulk lifetimes in CME sites around 20–25 s (*Picco et al., 2015*; *Sirotkin et al., 2010*). Single-molecule tracks of these proteins tagged with SNAP-SiR display short lifetimes averaging 1.4 to 1.6 s (*Figure 3* and *Table 2*), while Wsp1p displays even shorter lifetimes (mean = 1.15 s, 95%ile = 2.6 s). End4p and Clc1p, which have been reported to have bulk lifetimes around 40 s and 110 s (*Sirotkin et al., 2010*) also display short average single-molecule residence times of 1.6 and 1.5 s, respectively. All of the target proteins display a similar shape of residence time distribution, except for Fim1p, which contains a unique sub-population of short-lived events around 0.5 s. Small but statistically significant differences occur between the distributions of lifetimes for many samples (by Mann-Whitney test, *Figure 3—figure supplement 2*), with Fim1p and Wsp1p significantly different from all other samples (at p<0.0001). All samples except for Arc5p are significantly different from Pil1p (at p<0.05). All of the endocytic proteins we measured display much shorter lifetime than would be predicted for a simple assembly/disassembly model given their bulk lifetimes that have been reported previously (*Berro and Pollard, 2014a*; *Sirotkin et al., 2010*).

We attribute the disappearance of spots to dissociation of labeled proteins from endocytic sites, and not to photobleaching or other photophysical artifacts. We tested a range of tracking parameters to maximize the detected track lifetimes and minimize artifacts which might arise due to fluorophore blinking or missed detections in low signal/noise images (see Materials and methods and

**Table 3.** Statistics for single-molecule tracks of SNAP-tag fusion proteins imaged under alternate conditions.

| Name | Median, sec | Mean, sec | S.D., sec [*] | 95%ile, sec | N tracks | N movies (datasets with > 40 tracks) | N samples | D, nm²/sec [†] |
|------|------|------|------|------|------|------|------|------|
| Acp1p [high laser power] | 1.1 | 1.43 | 0.12 | 3.7 | 1968 | 11 (11) | 2 | 3367 |
| Acp1p [fixed][‡] | 12.0 | 15.3 | n.d. | 54.1 | 53 | 6 | 1 | n.d. |

[*] Standard Deviation is determined across means calculated from individual datasets (each movie), only counting movies with >40 tracks.

[†] Apparent diffusion coefficient calculated from mean squared-displacement over time for time windows < 1 s (10 points).

[‡] Acp1p-SiR sample fixed with formaldehyde and imaged under the same illumination conditions (low power) and camera settings as live samples (*Table 2*).

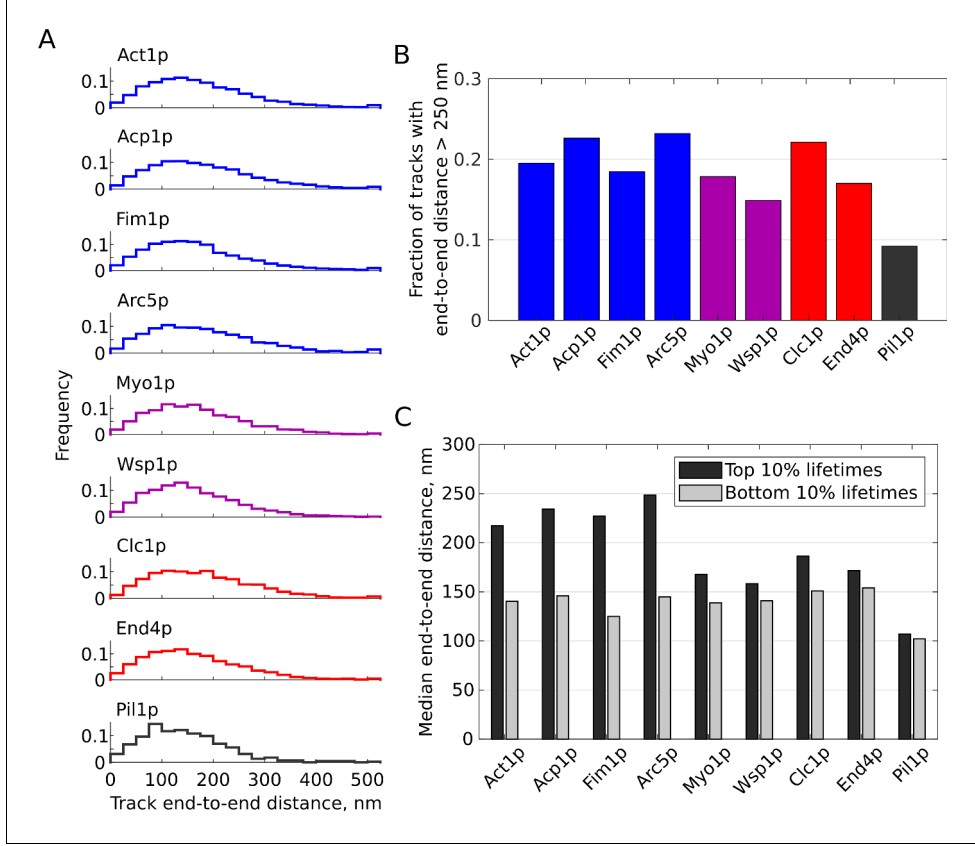

**Figure 4.** Single-molecule tracks net distances. (**A**) Distributions of end-to-end distances for tracks of SNAP-tag fusion proteins. Actin-associated endocytic proteins are shown in blue, nucleation-promoting factors in purple, membrane-associated endocytic proteins in red, Pil1p as an 'immobile' control in gray. (**B**) Fraction of tracks with end-to-end distance longer than 250 nm for each sample. Colors as in (**A**). (**C**) Median distances of tracks in the top or bottom 10% of lifetime distributions for each sample. *Figure 2—videos 6*, *7*, *Figure 3—videos 1*, *2*, *3*, *4*, *5*, *6*, *7* are representative movies from which the data shown in this figure have been extracted.

*Figure 3—figure supplement 3*). While increasing illumination intensity caused a corresponding increase in the global photobleaching rate as measured by the decay in the number of spots visible per frame (0.12 sec$^{-1}$ to 0.18 sec$^{-1}$), the distributions of track lifetimes did not change significantly (*Figure 3—figure supplement 1*, *Table 3*, *Figure 3—video 9*). Because a large fraction of the cell volume is illuminated, the reduced rate of appearance of new fluorescent molecules is a good approximation for the rate of photobleaching at the cell membrane. Additionally, in images of Acp1p-SiR cells fixed with paraformaldehyde, we could track many fluorescent spots over 15 s long under the same imaging conditions and recording settings (*Figure 3—figure supplement 1C*, *Table 3*, *Figure 2—video 7*). Therefore, while photobleaching does occur during the imaging time, it is about ten-fold slower than the spot lifetimes we measured and is not interfering with our ability to track single fluorophores.

## Characteristic motions of endocytic proteins are heterogeneous

The single-molecule tracks typically span a net distance of only 100 to 200 nm (*Figure 4A*), that is the distance traveled from the first appearance of a spot to its disappearance. Actin-associated proteins and clathrin contain a greater proportion of long tracks (>250 nm, *Figure 4B*). By two sample Mann-Whitney test, most of the differences between different proteins' distributions of track distances are statistically significant (at p<0.05, *Figure 3—figure supplement 2E*). Act1p is not significantly different from Myo1p, Myo1p is also not significantly different from Fim1p; Acp1p, Arc5p and Clc1p are not significantly different from each other; Wsp1p and End4p are not significantly different from each other; Pil1p is significantly different from all other samples. We also investigated the

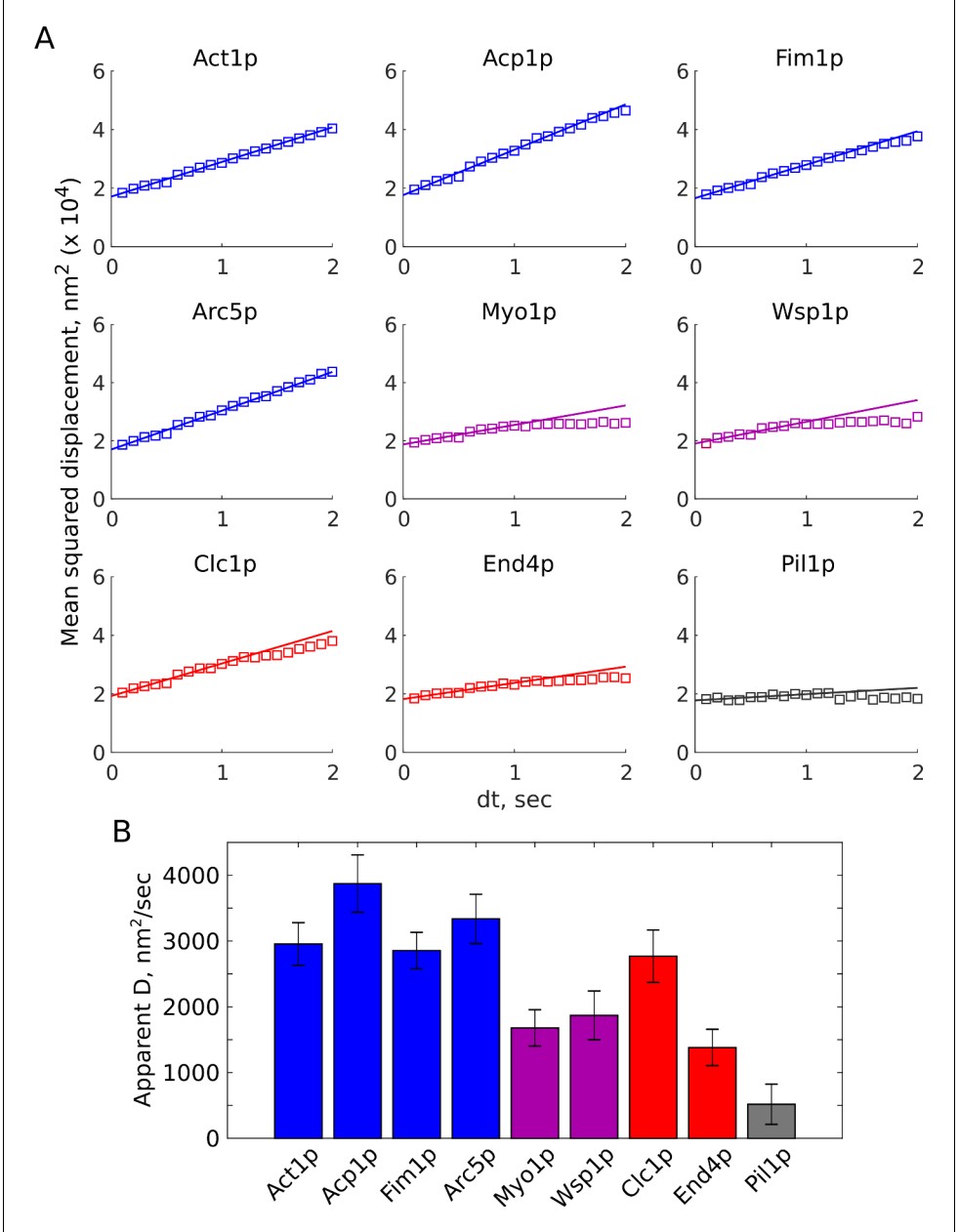

**Figure 5.** Average motions of endocytic proteins. (**A**) Mean squared-displacement over time, computed across all pairwise distances in all tracks (squares), with their linear fits (solid lines). (**B**) Apparent diffusion coefficients (*D*) for analyzed proteins, assuming 2-dimensional diffusion, *MSD = 4D\*dt + b*. Error bars show the 95% confidence interval associated with the fit of MSD vs. dt slope. Actin-associated endocytic proteins are shown in blue, nucleation-promoting factors in purple, membrane-associated endocytic proteins in red, Pil1p as an 'immobile' control in gray. *Figure 2—videos 6*, *7*, *Figure 3—videos 1*, *2*, *3*, *4*, *5*, *6*, *7* are representative movies from which the data shown in this figure have been extracted.

relationship between track lifetime and net distance traveled. For molecules undergoing pure diffusive motion, the net distance increases with the square root of the lifetime. However, we observed net distances depend on lifetime only for some proteins (*Figure 4C*). Comparing the tracks in the top and bottom 10% of lifetimes shows that long-lived tracks of actin-associated proteins (Act1p, Acp1p, Fim1p, and Arc5p) travel longer distance than short-lived tracks (*Figure 4B*). For membrane-bound endocytic proteins (Myo1p, Wsp1p, End4p, and to some extent, Clc1p) this dependence is

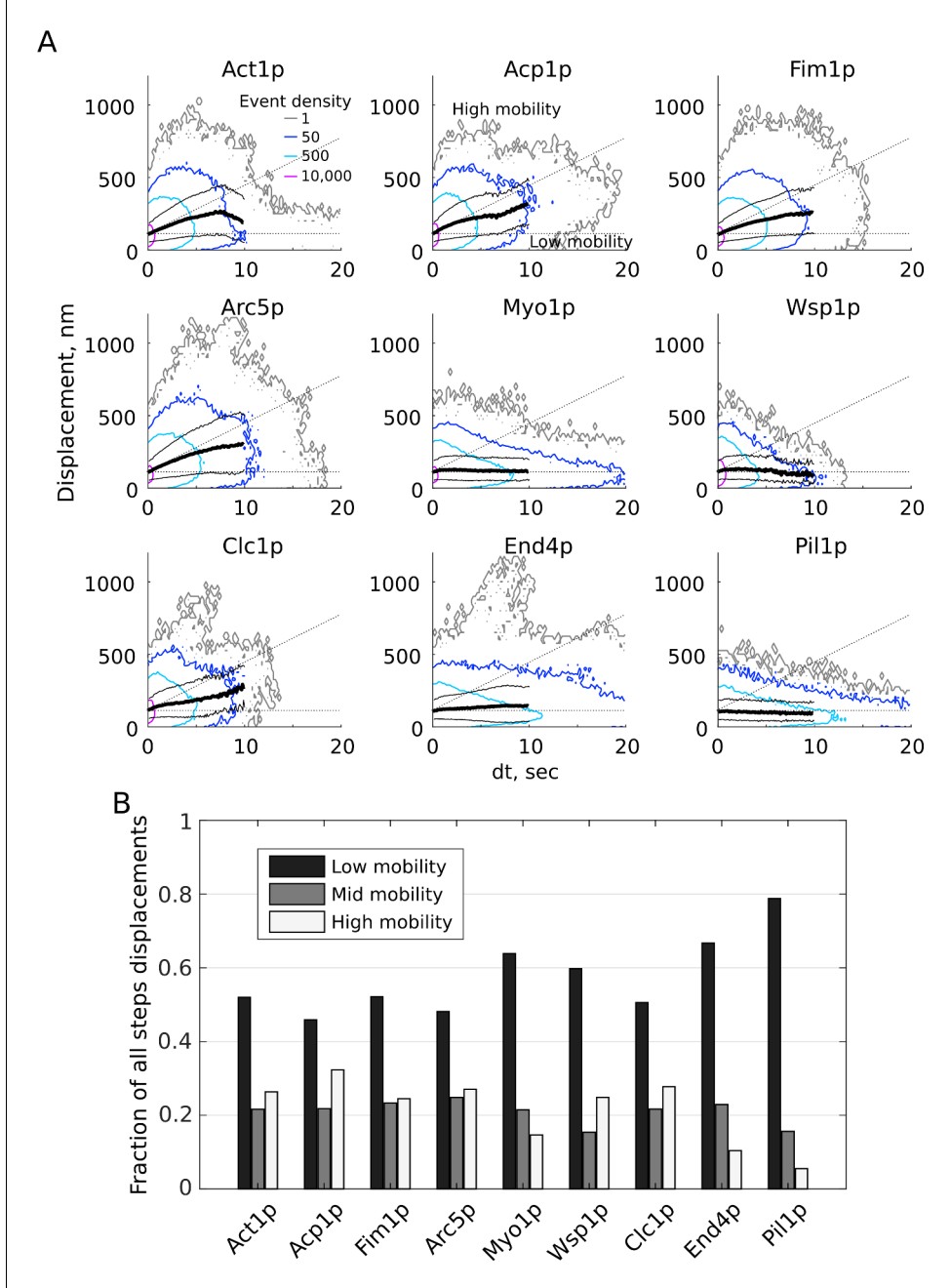

**Figure 6.** Distributions of single-molecule motions of endocytic proteins. (**A**) Pairwise distances over time calculated for all pairs of points in each track show a range of mobility behaviors. Contour lines represent 3D histogram bins of identical height, normalized by the total number of points in each dataset, with relative density (arbitrary scaling units as indicated in the first panel). The mean displacements over time (heavy black line) and standard deviation (thin black lines) are shown for each time step shorter than 10 s. The plots are divided into high-, mid-, and low-mobility zones as indicated by black dotted lines. (**B**) The fractions of all pairwise displacements from (**A**) that fall into high-, mid-, and low-mobility classes for each dataset. *Figure 2—videos 6*, *7*, *Figure 3—videos 1*, *2*, *3*, *4*, *5*, *6*, *7* are representative movies from which the data shown in this figure have been extracted.

lost (*Figure 4B*). Pil1p displays overall short displacements with a median track length of 134 nm, likely limited by the localization precision.

Next, we characterized each protein's average mobility by calculating the mean squared displacement (MSD) over time (*Figure 5*). For all the endocytic proteins the MSD is linear at short time scales, suggesting diffusive behavior, but appears sub-diffusive at longer timescales (not shown here, but see *Figure 6*). We used only the first 10 points (up to 1 s) for fitting, relying on the highest-confidence points and avoiding potential biases resulting from the spot localization precision and lower number of points at long time steps (*Michalet, 2010*), and avoiding the regime of sub-diffusive motion. The slopes of these curves were significantly different (*Figure 5A*), but the y-intercept values are related to the localization precision (*Michalet, 2010*) which was similar across all samples (*Figure 3—figure supplement 2A*). We computed the apparent diffusion coefficient D (*Figure 5B*), as $\langle dr^2 \rangle = 4D \cdot dt + b$, assuming a two-dimensional diffusion model because our images record only a single plane near the base of the cell, not three dimensions. Tracks of actin-associated proteins and clathrin have higher apparent mobility (2,700 to 3,800 nm$^2$/sec) than those of membrane-associated proteins Myo1p, Wsp1p, and End4p (1,300 to 1,800 nm$^2$/sec). However, these less-mobile endocytic proteins still display higher average mobility than Pil1p ($\sim$500 nm$^2$/sec), indicative of nanoscale motions within the CME membrane patch.

We observed a high degree of heterogeneity in the pairwise displacements (*Figure 6A*), that is all of the pairwise distances over all time points in each track. To identify classes of behaviors within these distributions, we divided each plot into areas of high, middle, and low mobility and determined the fraction of points in each category for each protein (*Figure 6B*). The cutoff between the high- and mid-mobility zones was set as the average behavior of actin-associated proteins, and the low-mobility cutoff of 125 nm corresponds to 2.5 times the average spot localization precision of 50 nm (*Figure 3—figure supplement 2*). Across the full datasets grouped in this way, high-mobility events have an average apparent D of 6,050 $\pm$ 720 nm$^2$/sec, mid-mobility events correspond to apparent D of 1,100 $\pm$ 14 nm$^2$/sec, and low-mobility events correspond to apparent D of 67 $\pm$ 35 nm$^2$/sec.

As expected, each protein sampled all three behaviors to some extent but membrane-associated proteins predominantly sample the low-mobility class. However, there are some notable exceptions and differences between proteins: Act1p contains a larger population of long-lived, low-mobility events than other actin-binding proteins; Clc1p and End4p contain a population of high-mobility displacements not observed in other membrane-bound proteins Wsp1p, Myo1p, and Pil1p. It is also important to compare the absolute displacements as well as the fractions of events in each class. For example, although Wsp1p has a similar fraction of high-mobility counts as Clc1p, Wsp1p does not contain the population of large-displacement (>500 nm) motions seen in Clc1p and other vesicle-associated proteins and instead the high-mobility events are almost exclusively comprised of short time steps (<300 nm and <1 s).

## Discussion

### Endocytic proteins display short residence times and rapid turnover

Our single-molecule tracking results indicate that actin and other endocytic proteins turn over multiple times during a CME event. This behavior was predicted based on modeling (*Berro et al., 2010*) and indirect measurements such as FRAP (*Kaksonen et al., 2003*; *Kaksonen et al., 2005*), but molecular residence times have never been directly observed in CME. We also confirmed the rapid turnover observed by our single-molecule approach with additional FRAP measurements of Wsp1p, Myo1p, Fim1p, and Acp1p (*Figure 2—figure supplement 1*). We report actin and associated endocytic proteins have residence times most often around 1 to 2 s, indicating the meshwork turns over up to five times during a single CME event.

Other actin structures such as lamellipodia generate sustained force by 'treadmilling', characterized by continuous assembly and disassembly and retrograde flow of monomers (*Pollard and Borisy, 2003*; *Smith et al., 2013*; *Watanabe and Mitchison, 2002*). Kaksonen and colleagues have used photobleaching experiments in mutant yeast strains which generate long actin comet-tails to determine that the polymerized actin at CME sites moves away from the cell membrane at approximately 50 nm/sec (*Kaksonen et al., 2003*), and single-molecule tracking measurements of actin in

lamellipodia report retrograde flow around 30 nm/sec (*Yamashiro et al., 2014*). Electron microscopy studies have shown that the size of the endocytic actin meshwork extends about 100 to 250 nm from the cell membrane (*Idrissi et al., 2012*; *Kukulski et al., 2012*) and mathematical modeling has shown that the endocytic actin filaments are typically 100 to 150 nm long (*Berro et al., 2010*). These findings, along with the partly overlapping presence of actin assembly and disassembly factors in endocytic patches, have suggested that actin monomers could travel the distance of the meshwork and turn over as rapidly as every 3 to 5 s (*Goode et al., 2015*). Consistent with those estimates and our own FRAP experiments, our single-molecule data are now the first direct observation that actin and associated proteins have typical single-molecule residence times around 1 to 2 s, and indeed rarely reside at an endocytic site longer than 3 s (*Figure 3*).

The characteristic shape of these residence time distributions, a peak with a skewed tail, indicates that the unbinding dynamics of these proteins do not follow a simple kinetic mechanism. Mass action kinetics predict an exponential distribution for single-step unbinding mechanism. In general, multi-step kinetic pathways result in residence times (or 'dwell times') following a gamma distribution (*Floyd et al., 2010*). Correspondingly, the residence times of actin monomers in a treadmilling filament have been shown to follow complex kinetics resulting in a peak shape that depends on all of the rates of polymerization, aging, cofilin binding, and severing steps (*Roland et al., 2008*). Although we have not extracted specific rate constants from our data, the shapes of the residence time distributions appear similar to those models. The skewed peak shape also argues against the interpretation that these single-molecule events arise from transient associations with stable assemblies, which would instead result in exponential dwell time distributions by mass-action kinetics.

The dissociation kinetics for actin-associated proteins are likely driven by the disassembly and severing of the actin filaments to which they are bound rather than simple unbinding, because the unbinding rates measured for these proteins in vitro are typically far slower than the timescales we observed here. For example, unbinding rates from actin filaments have been reported for Acp1p in the range of $10^{-4}$ sec$^{-1}$ (*Bombardier et al., 2015*; *Kuhn and Pollard, 2007*) and for Fim1p in the range of $10^{-2}$ to $10^{-3}$ sec$^{-1}$ (*Skau et al., 2011*). The slight differences between residence times of actin-associated proteins may reflect the delay of binding a filament only after its nucleation (as Fim1p and Acp1p residence times are slightly shorter than Act1p), or the additional time for pre-recruitment before nucleation of the actin branch (as Arc5p residence times are slightly longer than Act1p). The fast sub-population of Fim1p tracks around 0.5 s may correspond to transient binding events that engage one but not both actin-binding domains before unbinding; such a prominent population of transient events is absent in other proteins.

The similar shape and timescale for residence times of membrane-associated proteins might be explained by other unique combinations of rates governing those proteins' interactions with the membrane and many binding partners, especially in oligomeric lattice assemblies such as the clathrin coat. The short lifetimes we measured for clathrin (Clc1p) agree with previous reports that individual clathrin triskelia unbind during CCP development in order to achieve the necessary change in curvature (*Avinoam et al., 2015*). Previous studies also reported that WASp rapidly exchanges at sites of actin nucleation (*Weisswange et al., 2009*), although these dynamics have not been measured directly in CME. While both End4p and Clc1p are known to reside at endocytic sites long before CCP invagination and scission (bulk lifetimes around 40 s and 110 s, respectively *Sirotkin et al., 2010*), we observe considerably fewer long-lived immobile events for Clc1p than for End4p (*Figure 6*), suggesting that clathrin turns over throughout the CCP lifetime and not exclusively during late stages of membrane invagination. Consistent with its proposed role in anchoring the actin meshwork with the CCP membrane, End4p (Sla2 in budding yeast) displays the highest fraction of long-lived events compared with other endocytic proteins.

## Motions of endocytic proteins are heterogeneous

The high-, mid-, and low-mobility behaviors shown in *Figure 6* (average apparent D for each class ~6,000 nm$^2$/sec, ~1,100 nm$^2$/sec, ~65 nm$^2$/sec) are consistent with vesicle diffusion, motions within the CCP-associated meshwork, and membrane-bound states, respectively. We attribute the longer distances traveled by some tracks of actin-associated proteins (*Figure 4C*, *Figure 6*) to be molecules that remain associated with the endocytic vesicle after scission. However, the large majority of events have very short end-to-end distance (*Figure 4A–B*). This apparent over-representation of events with restricted motions is consistent with a switch in behavior of the CME assembly from

rapid turnover with small motions in the developing CCP to reduced recruitment and sustained disassembly around the vesicle after scission. Unique sub-populations are apparent in several proteins' behaviors (*Figure 6*), suggesting various motions for long- or short-lived components of the CME meshwork. For all proteins, very long-timescale events (>10 s) generally exhibit only short displacements, indicating that these molecules are restricted to small motions on the CCP membrane and are disassembled from the diffusing CCV relatively quickly.

## Limitations to current method and data

Single-molecule tracking is a powerful technique to uncover molecular mechanisms in complex systems. We acknowledge that the results of single-molecule tracking are sensitive to the chosen tracking parameters (*Jaqaman et al., 2008*; *Smith et al., 2011*), especially because the images' low signal-to-background ratio results in some missed detections of spots. However, even across a broad range of parameters up to unrealistic values that introduce verifiable artifacts, the average single-molecule residence times remain below 3 to 4 s (*Figure 3—figure supplement 3*), still much faster than the ~10 s prediction for single-turnover mechanisms (*Figure 1*).

Most of the proteins we tracked have already been reported to specifically localize to CME patches (*Goode et al., 2015*) and therefore we did not apply any filtering to remove non-endocytic fluorescent tracks. We considered the possibility that some long-lived, low mobility actin tracks might correspond to structures such as actin cables, cytokinetic rings, or unproductive clathrin patches, but the absence of such events in Acp1p and Fim1p argue against that interpretation. Additionally, fluorescent events which persisted longer than 40 s were excluded from analysis as potential artifacts from tracking these or other non-endocytic structures. Due to the various orientations of CCP motions at the cell tips, sides and base, and our localization precision of ~50 nm we did not attempt to align trajectories or perform more detailed analyses of motions within the endocytic structures, but such heterogeneities may be addressable with future imaging studies.

## Implications of actin turnover for force production models

Our results have important qualitative and quantitative implications for models of the endocytic machinery. They support the idea that the actin network can remodel during CME and generate greater total force than has been previously estimated. Previous discussions of CME often assumed that the entire actin meshwork and membrane coat assembles and disassembles once during the total lifetime of the patch, with 'turnover' referring to the release of proteins to be re-used in a later CME event. Discrete phases of assembly and disassembly of actin monomers (and associated proteins) would result in an average residence time of ~10 s (*Figure 1*), inconsistent with our results. Although some earlier reports have suggested the existence of turnover within the CME machinery, current literature still generally assumes the endocytic meshwork assembles and disassembles in discrete phases (*Carlsson, 2018*; *Hassinger et al., 2017*; *Mund et al., 2018*; *Picco et al., 2018*; *Wang and Carlsson, 2017*).

An actin and membrane-protein assembly undergoing rapid and continuous turnover will behave differently from a sequentially assembled and disassembled structure in several ways. If more total molecules of actin are consumed by the endocytic meshwork, the total chemical energy of polymerization released through ATP hydrolysis must be several times higher than has been previously estimated. To accumulate the known peak number of actin molecules while sustaining such a high rate of turnover (consuming up to five times more actin), the rates of polymerization and disassembly must both be higher than previously estimated, and turnover of the actin-nucleation promoting factors Wsp1p and Myo1p would generate a higher number of polymerizing ends over the lifetime of the CME event. The endocytic patch contains approximately 6,500 actin monomers at its peak (*Sirotkin et al., 2010*) MBoC, Berro 2010 MBoC), but based on the residence times we report here, a single patch might consume more than 30,000 actin subunits over time. A recent report proposed that myosin-Is (Myo3 and Myo5 in budding yeast) enable higher rates of actin polymerization by pushing the filament ends away from the membrane (*Manenschijn et al., 2019*). Another recent report demonstrated that mechanical stress in highly-crosslinked actin networks can greatly enhance the rate of ADF/cofilin severing of actin filaments (*Wioland et al., 2019*), which could also contribute to the high turnover rates we observe in CME patches. Fast turnover of the actin meshwork components could affect other higher-order mechanisms of force generation. For example, continuous

unbinding of fimbrin crosslinkers would allow remodeling of the meshwork and spatiotemporal redistribution of elastic energy stored in crosslinkers (*Ma and Berro, 2018*; *Ma and Berro, 2019b*; *Picco et al., 2018*). Such remodeling of the actin meshwork could also enable important changes in the orientation of filaments and their forces over time. While our paper was in preparation, *Nickaeen et al. (2019)* reported simulations based on the dendritic nucleation model, which requires actin disassembly and turnover, indicating that the endocytic actin meshwork can sustain forces sufficient to achieve the membrane invagination. Our results are a key experimental validation of this predicted turnover behavior, which was not directly accessible in previous experimental and theoretical studies.

Future studies of CME should consider the endocytic actin meshwork and membrane coat as highly dynamic, turning over multiple times to generate, distribute, and transmit the forces that deform the plasma membrane into a vesicle. Novel experimental methods will be needed to measure rates and identify specific molecular complexes at the endocytic site. We expect that the increasing accessibility of super-resolution microscopy and single-molecule techniques as well as added levels of detail in computational models will enable further exploration of the molecular mechanisms of CME and other complex molecular assemblies in live cells.

# Materials and methods

## Key resources table

| Reagent type (species) or resource | Designation | Source or reference | Identifiers | Additional information |
|---|---|---|---|---|
| Strain, strain background (*Schizosaccharomyces pombe*) | Wild-type | S Forsburg | FY527 | *ade6-M216 his3-D1 leu1-32 ura4-D18 h-* |
| Strain, strain background (*S. pombe*) | Fim1p-SNAP | This study | JB135 | *fim1-SNAP-kanMX6 ade6-M216 his3-D1 leu1-32 ura4-D18 h-* |
| Strain, strain background (*S. pombe*) | Acp2p-mEGFP/ Fim1p-SNAP | This study | JB150 | *acp2-mEGFP-kanMX6 fim1-SNAP-kanMX6 ade6-M216 his3-D1 leu1-32 ura4-D18 h-* |
| Strain, strain background (*S. pombe*) | Pil1p-SNAP | (*Lacy et al., 2017*) | JB198 | *pil1-SNAP-kanMX6 ade6-M216 his3-D1 leu1-32 ura4-D18 h-* |
| Strain, strain background (*S. pombe*) | Clc1p-SNAP | This study | JB202 | *clc1-SNAP-kanMX6 ade6-M216 his3-D1 leu1-32 ura4-D18 h-* |
| Strain, strain background (*S. pombe*) | SNAP-Act1p | This study | JB216 | *41nmt1-SNAP-actin-leu+ ade6-M216 his3-D1 ura4-D19 h-* |
| Strain, strain background (*S. pombe*) | SNAP-Myo1p | This study | JB304 | *SNAP-myo1 fex1Δ fex2Δ ade6-M216 his3-D1 leu1-32 ura4-D18 h-* |
| Strain, strain background (*S. pombe*) | Acp1p-SNAP | This study | JB305 | *acp1-SNAP fex1Δ fex2Δ ade6-M216 his3-D1 leu1-32 ura4-D18 h-* |
| Strain, strain background (*S. pombe*) | End4p-SNAP | This study | JB307 | *end4-SNAP fex1Δ fex2Δ ade6-M216 his3-D1 leu1-32 ura4-D18 h-* |
| Strain, strain background (*S. pombe*) | Arc5p-SNAP | This study | JB346 | *arc5-SNAP fex1Δ fex2Δ ade6-M216 his3-D1 leu1-32 ura4-D18 h-* |
| Strain, strain background (*S. pombe*) | Acp1p-mEGFP | This study | JB366 | *acp1-mEGFP fex1Δ fex2Δ ade6-M216 his3-D1 leu1-32 ura4-D18 h-* |
| Strain, strain background (*S. pombe*) | SNAP-Wsp1p | This study | JB393 | *SNAP-wsp1 fex1Δ fex2Δ ade6-M216 his3-D1 leu1-32 ura4-D18 h-* |
| Strain, strain background (*S. pombe*) | mEGFP-Wsp1p | This study | JB385 | *mEGFP-wsp1 fex1Δ fex2Δ ade6-M216 his3-D1 leu1-32 ura4-D18 h-* |
| Strain, strain background (*S. pombe*) | Acp1p-mEGFP/Fim1p-mCherry | (*Berro and Pollard, 2014b*) | JB54 | *acp1-mEGFP-kanMX6 fim1-mCherry-NatMX6 ade6-M216 his3-D1 leu1-32 ura4-D18 h+* |
| Strain, strain background (*S. pombe*) | Fim1p-mEGFP | (*Berro and Pollard, 2014b*) | JB57 | *fim1-mEGFP-NatMX6 his3-D1 leu1-32 ura4-D18 leu1-32 h+* |

*Continued on next page*

*Continued*

| Reagent type (species) or resource | Designation | Source or reference | Identifiers | Additional information |
|---|---|---|---|---|
| Strain, strain background (*S. pombe*) | mEGFP-Myo1p | (*Sirotkin et al., 2005*) | TP195 (JB205) | *kanMX6-Pmyo1-mGFP-myo1 leu1-32 ura4-D18 his3-D1 ade6 h-* |
| Other | SNAP-Cell 647-SiR | New England Biolabs | S9102S | Dissolved to 100 µM in DMSO and aliquoted, stored −80°C |
| Software, algorithm | Matlab | Mathworks | | Scripts are included as Source code |
| Software, algorithm | FIJI ImageJ | (*Schindelin et al., 2012*; *Schneider et al., 2012*) | | |
| Software, algorithm | Python Microscopy Environment (PYME) | www.python-microscopy.org; (*Baddeley et al., 2011*) | | |

## Model simulations

Simulated profiles of numbers of molecules over time were calculated with Matlab (Mathworks), as lists of arrival and departure timestamps generated according to each type of logical model. The simulation script is given as *Source code 1*.

For 'Random assembly' models, arrival times were randomly chosen according to a normal distribution centered at one quarter of the total patch lifetime and standard deviation of one tenth of the total lifetime. For 'Random assembly/Random disassembly', the departure times were randomly chosen according to a normal distribution centered at three quarters of the total lifetime and standard deviation of one tenth of the total lifetime. For 'Random assembly with First-in/First-out disassembly', arrival and departure times were chosen on the same distributions as above, but were sorted so that the corresponding values of binding times and unbinding times were matched in order from lowest-to-highest. Residence times were then calculated by subtracting each unbinding time from its corresponding binding time.

For 'Continuous turnover' model, arrival times were randomly chosen according to a broader normal distribution, taking the absolute value of a distribution centered at one third of the total lifetime and standard deviation of one fifth of the total lifetime. Residence times were chosen according to a gamma distribution with shape parameter equal to 2 and scale parameter equal to 1, to resemble a multi-step dwell time kinetic giving a shape and mean similar to our observed lifetime distributions. Departure times were calculated by taking the sum of the binding time and a residence time selected at random from this distribution.

The profile of number of molecules present in the patch at each 1 s time bin was calculated as the number of molecules with arrival time smaller than or equal to the current time bin and departure time larger than the current time bin. Simulations were performed with 900 total molecules in a lifetime of 20 s, repeated 50 times for each model.

## Yeast strains

We generated *Schizosaccharomyces pombe* strains with SNAP-tag (Addgene plasmid 87024) or mEGFP (Addgene plasmid 87023) inserted at the genomic loci of various proteins, using either homologous recombination with kanamycin selection (*Bähler et al., 1998*) or CRISPR-Cas9 gene editing with fluoride selection (*Fernandez and Berro, 2016*). Because *S. pombe* is inviable when its sole source of actin is fused with a fluorescent protein (*Wu et al., 2008*; *Wu and Pollard, 2005*), we integrated SNAP-act1 into the *leu1+* locus under control of the *41nmt* promoter, as previous studies have done for mEGFP-act1. We did not measure the expression level of SNAP-Act1p or its effect on actin functionality, but we expect it to behave similarly to mEGFP-Act1p used previously. Previous studies using this strategy report the actin fusion protein represents around 5% of the total actin in the cell (*Berro and Pollard, 2014a*; *Sirotkin et al., 2010*; *Wu et al., 2008*; *Wu and Pollard, 2005*). The strains used in this study are listed in the Key Resources Table.

## Growth and SNAP-tag labeling

Cells were grown in liquid YE5S medium at 32°C to exponential phase (OD$_{595}$ between 0.4 and 0.6) then diluted into liquid EMM5S medium and grown for 12 to 24 hr at 25°C before labeling with

SNAP fluorophore (*Keppler et al., 2003*; *Lukinavičius et al., 2015*; *Stagge et al., 2013*). As discussed previously (*Lacy et al., 2017*), the SNAP-substrate fluorophore does not accumulate in high amounts in yeast cells containing multidrug exporter genes and intact cell walls. Cells were diluted to 0.1 $OD_{595}$ in 1 mL of EMM5S containing 1 µM of silicon-rhodamine benzylguanine derivative SNAP-SiR647 (SNAP-Cell 647-SiR, New England Biolabs). Culture tubes were wrapped in aluminum foil to protect them from light and incubated on a rotator overnight (about 15 hr) at 25℃. Cells were washed five times by centrifuging at 1500xg for 3 min and resuspending in 1 mL of EMM5S, then incubated at 25℃ for an additional hour, then washed five times again by centrifuging at 1500xg for 3 min and resuspending in 1 mL of EMM5S. Cells were finally resuspended in 20 to 100 µL of 0.22 µm filtered EMM5S to achieve suitable density for imaging. SNAP-tag labeling efficiency was not determined before tracking analysis, but is estimated to be around 0.1% to 1% of the SNAP-tag fusion protein's expression level.

Cells expressing mEGFP fusion were grown in YE5S and EMM5S liquid media as above, then washed once by centrifuging at 1500xg for 3 min and resuspending in 20 to 100 µL of 0.22 µm filtered EMM5S to achieve suitable density for imaging.

## Paraformaldehyde fixation

For imaging fixed cells, SNAP-tag expressing cells were grown and labeled and washed as above. Cells were incubated in 3.6% paraformaldehyde solution for 15 min on a rotator at room temperature. Cells were then pelleted by centrifugation at 10,000xg for 1 min and washed three times by resuspending in filtered EMM5S and centrifuging at 1500xg for 3 min. Cells were finally resuspended in 20 to 100 µL of 0.22 µm filtered EMM5S to achieve suitable density for imaging.

## Imaging

Cells were pipetted on to pads of 25% gelatin prepared with 0.22 µm filtered EMM5S, covered with a #1.5 coverslip, which had been washed in ethanol for 30 min and plasma cleaned for 3 min to avoid nonspecific dye or other autofluorescent particles on the surface, and sealed around the edges with Valap. Samples were imaged on an Eclipse Ti inverted microscope (Nikon) equipped for through-objective TIRF, with a 642 nm excitation laser for SiR imaging (Spectral Applied Research) and a 488 nm laser for mEGFP imaging (Spectra-Physics). Images were recorded through a 60x/1.49 NA Apo TIRF objective (Nikon) and further magnified with the microscope's 1.5x lens, and detected using an iXon DU897 EMCCD camera (Andor); image pixels correspond to 178 nm. The microscope, camera, and illumination were controlled through Nikon Elements software.

For single-molecule tracking in SNAP-SiR labeled samples of live or fixed cells, the imaging focal plane is set about 1 to 1.5 µm below the cell midplane, just above the base of the cells adjacent to the coverslip, and the laser is angled for near-TIRF illumination, so that spots at the membrane are in focus and cytoplasmic fluorescence is reduced. The 642 nm laser illumination intensity was 0.5 or 0.8 $W/cm^2$ (measured exiting the objective). The camera was set to 100 msec exposure, with EM gain set to 300 at 5 MHz readout with 14-bit digitization depth. We recorded 60 s movies, starting recording with the laser off and turning it on after a few frames so that the initial fluorescence signal is not lost due to hardware delay times.

For endocytic patch tracking in samples expressing mEGFP, the 488 nm laser illumination intensity exiting the objective was 0.2 $W/cm^2$, with camera exposure 100 msec and EM gain 300. We recorded 120 s movies, starting with the laser off and turning it on after a few frames.

For two color imaging, we collected alternating red-channel and green-channel images, switching between 642 nm laser (0.8 $W/cm^2$) and 488 nm laser (0.2 $W/cm^2$) with 200 msec exposure time and EM gain set to 300, with about 1 s delay between acquisition frames to switch filter sets. A sample of Tetraspeck beads (0.2 µm) was prepared and imaged using the same protocol to ensure alignment between channels.

## Fluorescence Recovery after Photobleaching experiments

Cells were grown in filtered EMM5S for at least 24 hr before imaging. We used an SP8 laser scanning confocal microscope (Leica) with a 100x/1.44 NA HC PL APO objective (Leica). We used a 488 nm Argon laser (LASOS) to image (illumination intensity exiting the objective for Fim1p-mEGFP: 0.29 $mW/cm^2$, 563.7 V PMT gain, other samples: 0.56 $mW/cm^2$, 725.5 V PMT gain) and to bleach

regions of interest (illumination intensity 2.62 mW/cm$^2$ using FRAP Booster and Zoomer functions). We imaged a single focal plane (pinhole of 1 Airy unit, i.e. 147.4 µm) at the surface of yeast cells closest to the coverslip (dimensions 512 × 64 pixels) zoomed 2.26x at 700 Hz using an average of two line scans in bidirectional mode at 10 frames per second; pixel size corresponds to 100.7 nm. Imaging and bleaching were controlled using the Leica Application Suite X (v 3.5.2.18963) software.

For each field of view, we bleached a region of interest for three to ten 100 ms frames (depending on strain and experiment) with 10 s imaging before and after the bleaching pulse. The bleaching region covered several cells and was perpendicular to their long axis (surface area around 1.5 µm x 3.5 µm) in order to limit bleaching of the rest of the cell.

Movie analysis and spot tracking were performed using the Fiji distribution of ImageJ (v 1.52 p) (*Schindelin et al., 2012*; *Schneider et al., 2012*) and the PatchTrackingTools plug-in (*Berro and Pollard, 2014a*; *Lemière and Berro, 2018*). In brief, each spot was circled with a 7-px circular region of interest and followed over time. We estimated background fluorescence around each spot using a 9-px median filter, which we subtracted from the spot integrated intensity. We corrected intensities for photobleaching by estimating the photobleaching rate from areas of cytoplasm void of endocytic patches outside the bleached regions.

## Image analysis – single molecule localization and tracking

Super-resolution spot localization and track generation were performed in the Python Microscopy Environment (PYME) (http://www.python-microscopy.org; *Baddeley et al., 2011*). Metadata for the images were determined as follows: camera read noise (88.8), from manufacturer's specifications; camera noise factor (1.41), standard correction for EMCCD cameras; electrons per count (49) and true EM gain (167), by calibration of blank movies recorded at varying intensity and gain settings, following *Hirsch et al. (2013)*; camera AD Offset (105), by measuring the average pixel intensity of dark frames before activating the laser. Each movie was processed for spot detection using a 2D Gaussian model with local intensity scaling threshold factor of 0.8; 'debounce rad', or the distance (in pixels) within which two fluorophores cannot be distinguished as 3; and the temporal background subtraction feature turned off. The local background and total spot intensity are both fitted with 2D Gaussians and subtracted to determine the spot intensity.

Candidate spots were filtered based on spot intensity (20 to 200 AU), spot size sigma (100 to 300 nm), and localization precision in x and y (0 to 150 nm). Single-molecule trajectories were generated by linking spots in successive frames within a limit of 100 nm, and allowing a maximum gap of 6 frames (spanning missed localizations in up to four frames), as discussed in the Materials and methods. Any spots which are not incorporated into a track of at least three spots, and any tracks which appear outside the cells, are discarded. Localization and tracking results were exported as text files to be further analyzed in Matlab.

Spots in the fixed sample of Acp1p-SiR were analyzed using the TrackMate plugin (*Tinevez et al., 2017*) for FIJI (*Schindelin et al., 2012*; *Schneider et al., 2012*), using spot detection with the Difference of Gaussians detector, threshold 2.5, maximum diameter 1 µm, and median filtering, and tracked using semi-automated manual tracking. We did not apply strict limits to spot quality and signal thresholds but rather relied on visual inspection and automated detection of spots' signal in successive frames, allowing gaps of missed localizations up to four frames.

## Image analysis – GFP patch tracking

We tracked GFP spots with the TrackMate plugin (*Tinevez et al., 2017*) for FIJI (*Schindelin et al., 2012*; *Schneider et al., 2012*), using spot detection with the Difference of Gaussians detector, threshold 2.0, maximum diameter 1 µm, and median filtering, and tracked using semi-automated manual tracking. We visually confirmed that tracks did not overlap with neighboring spots and spanned the full lifetime of intensity increase and decay, rather than a sharp disappearance due to diffusing out of the focal plane. Only these manually-curated tracks were exported for further analysis in Matlab.

Spots in the two-color images were identified and scored manually. Spot detection and colocalization algorithms would have difficulty because the GFP patches vary widely in intensity.

## Data analysis

Tracking data from PYME were read, curated, and analyzed with custom-written code in Matlab (Mathworks, *Source code 2* and *3*). We excluded tracks with lifetime shorter than 0.3 s or longer than 40 s and tracks which began before the start-time cutoff determined for each protein (see *Figure 3—figure supplement 4*). Because the first few frames of the movies are more crowded (potentially introducing errors in track linking) and may contain molecules already present in CME structures (yielding only partial information on the molecule's true trajectory) or other artifacts (e.g. high background or molecular aggregates that have not been photobleached), tracks which started within the cutoff time after the laser was turned on were discarded. Tracks longer than 40 s were attributed to artifacts or non-endocytic structures and also discarded. We chose the appropriate start-time cutoffs for each protein dataset as the first time point where the difference between the current lifetimes distribution and the distribution using the next cutoff (1 s later) was not significant by KS test.

A variety of features are calculated from the timestamp and position data such as the lifetime, stepwise displacements, velocity, and other derivative characteristics at the level of single spots or tracks (*Source data 1*). Statistics for numbers of tracks, movies analyzed, and independent samples prepared are given in *Table 2*. Single-molecule tracking datasets for endocytic proteins typically represent 10 to 30 movies analyzed from between two to five independent samples. Standard deviation of single-molecule residence time data was determined by calculating the mean track lifetimes from each movie individually, and computing the standard deviation of these separate results. Where indicated, we used a two-sample Mann-Whitney test for statistical significance of differences between distributions.

## Characterization of tracking parameters and data quality

Features of the spot localizations and tracking statistics are given in *Figure 3—figure supplements 2* and *3*. The low signal-to-noise ratio in single molecule images causes some spots to fall below quality thresholds for the detection algorithm, introducing the possibility of errors in tracking (*Jaqaman et al., 2008*; *Smith et al., 2011*). To determine an optimal set of analysis parameters that would both maximize successful detections and minimize false spots and incorrect track links, we generated tracks using a broad range of tracking parameters (gap-closing limit) for three representative proteins, Acp1p, Wsp1p, and Pil1p (*Figure 3—figure supplement 3*). Across this range of parameters, we observe small changes to the absolute values of the track lifetimes and a conservation of the trends between different proteins (*Figure 3—figure supplement 3A*). With very short linking limit, we observe many cases where a single spot is represented by multiple successive short tracks at the same position, but also an overall reduction in the number of tracks generated (*Figure 3—figure supplement 3B*). With very long linking limits, the gap-time distribution deviates from its exponential shape (*Figure 3—figure supplement 3D*), and the apparent residence time increases with labeling density (*Figure 3—figure supplement 3E*), suggesting that these excessively long gaps are linking false spots arising from cytoplasmic background or linking new labeled molecules that arrive at the site instead of simply accounting for missed detections of the same molecule. For Pil1p, where new labeled molecules specifically bind to the same sites over time (*Lacy et al., 2017*), the number of detected tracks plateaus and decreases with over-linking (*Figure 3—figure supplement 3B*). Even when setting an extremely long gap-closing limit (20 frames, or 2 s), which clearly introduces incorrect links between isolated events, the average residence times of endocytic proteins follow broadened distributions with averages below 4 s. Based on these observations, we chose the linking limit of 6 frames (including the frame before and after the gap), which allows a maximum of 4 consecutive missed detections (0.4 s, occurring for about 5% of all gaps, *Figure 3—figure supplement 2B*).

To avoid the possibility of multiple labeled proteins in the same endocytic patch or multiple spots overlapping in crowded regions of the cell, we chose cutoff times for each target protein to allow for an initial 'dilution' of the effective label density by photobleaching. Indeed, we observed in some samples that spots present at the start of recording are longer-lived than spots that appear later (*Figure 3—figure supplement 4*), presumably because the relatively higher label density causes neighboring spots to merge or an endocytic site to contain more than one fluorophore, or because of non-endocytic aggregates of molecules. Therefore, tracks that started in the initial frames were

excluded from analysis (typically 1 s, or longer for high copy number target proteins such as Fim1p or Pil1p), so that typically 2 to 3 molecules or fewer are visible per cell at any single time. We note that our current tracking methods are improved over those used in *Lacy et al. (2017)*, enabling us to include short-lived events (minimum lifetime of 0.3 s) and observe the non-exponential shape of Pil1p-SiR residence times.

Because the samples were imaged in a single focal plane near the base of the cells using partial-TIRF, we do not have direct access to the 3D motion behavior of molecules along the axis of CCP internalization. In the current dataset, we primarily aimed to investigate track lifetimes and so we imaged under very low illumination power and did not apply stringent filters on spot localization precision. Future studies, perhaps imaging under different illumination conditions, could aim to collect higher-resolution data to investigate spatiotemporal dynamics in greater detail.

## Software and reagents availability

The Python microscopy environment (PYME) software is available at www.python-microscopy.org, and our Matlab analysis script and tracking analysis data files are available as Source Code Files of this paper. We will happily share any *S. pombe* strains upon request and plasmids are available through Addgene.

## Acknowledgements

We thank Ronan Fernandez for assistance in creating yeast strains, Olivier Trottier for assistance with an earlier version of the Matlab analysis scripts, and members of the Berro lab for helpful discussions. We thank Joerg Nikolaus and the West Campus Imaging Core (Leica Center of Excellence) at Yale University for providing access to the Leica SP8 and help with FRAP experiments. This research was supported by National Institutes of Health/National Institute of General Medical Sciences Grant R01GM115636. MML was supported by National Institutes of Health Training Grant T32GM008283. We also acknowledge support from the Yale Program in Physics Engineering and Biology.

## Additional information

### Funding

| Funder | Grant reference number | Author |
|---|---|---|
| National Institute of General Medical Sciences | R01GM115636 | Michael M Lacy<br>Julien Berro |
| National Institute of General Medical Sciences | T32GM008283 | Michael M Lacy |
| Yale University | Program in Physics Engineering and Biology | Michael M Lacy<br>David Baddeley<br>Julien Berro |

The funders had no role in study design, data collection and interpretation, or the decision to submit the work for publication.

### Author contributions

Michael M Lacy, Conceptualization, Software, Formal analysis, Investigation, Visualization, Methodology; David Baddeley, Resources, Software, Supervision, Methodology; Julien Berro, Conceptualization, Resources, Formal analysis, Supervision, Funding acquisition, Investigation, Visualization, Methodology

### Author ORCIDs

Michael M Lacy https://orcid.org/0000-0003-0498-2817
Julien Berro https://orcid.org/0000-0002-9560-8646

### Decision letter and Author response

Decision letter https://doi.org/10.7554/eLife.52355.sa1

Author response https://doi.org/10.7554/eLife.52355.sa2

## Additional files

### Supplementary files

• Source code 1. Matlab simulation script for modeling assembly and disassembly profiles to illustrate the logic of 'turnover'.

• Source code 2. Matlab analysis scripts for reading track files and processing data for analysis of lifetimes, motions, and other features. This script requires raw tracking data (from PYME or similar format) as input files.

• Source code 3. Matlab analysis scripts for compiling and plotting lifetimes data from track analysis results. This script requires the output of process_tracks.m as an input (i.e. SuppFile4sets.mat).

• Source data 1. Matlab data structure containing the analyzed tracking data (lifetimes, positions, etc.) for tracks, the output of *Source code 2* (SuppFile2rocess_tracks.m) after filtering for start time cutoffs. The data from this file can be extracted using *Source code 3* (SuppFile3multiplot_dwell-times.m) and further processed to generate all the figures and calculations presented in the manuscript.

• Transparent reporting form

### Data availability

All data generated or analysed during this study are included in the manuscript and Source data 1 (Matlab data file).

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
