## [Decision Letter]

**Acceptance summary:**

Coordinated polymerization of actin filaments provides forces for plasma membrane deformation during endocytosis. However, the dynamics of individual actin molecules and actin-associated proteins within endocytic sites have not been reported, and thus the precise mechanism by which actin dynamics produces force for membrane deformation in endocytosis is incompletely understood. Here, Lacy et al. applied single-molecule speckle tracking to analyze the turnover of actin and actin-associated proteins in endocytic patches of fission yeast. They revealed very rapid turnover of molecules within the endocytic actin network, and heterogeneous behaviors of these proteins at the molecular level. Together, these results provide important new insights into the mechanism by which actin dynamics contributes to clathrin-mediated endocytosis.

**Decision letter after peer review:**

Thank you for submitting your work entitled "Single-molecule turnover dynamics of actin and membrane coat proteins in clathrin-mediated endocytosis" for consideration by *eLife*. Your article has been reviewed by three peer reviewers, one of whom is a member of our Board of Reviewing Editors, and the evaluation has been overseen by a Senior Editor. The reviewers have opted to remain anonymous.

Actin dynamics provides forces for plasma membrane deformation during endocytosis. Here, Lacy et al. applied single-molecule speckle tracking to analyze the turnover of actin and actin-associated proteins in endocytic patches of fission yeast. The revealed very rapid turnover of molecules within the endocytic actin network, and heterogeneous behaviors of these proteins at the molecular level. Thus, the authors suggest that the forces produced through actin polymerization in endocytic patches may be higher than previously estimated.

All reviewers concurred that the findings presented are potentially important. However, they stated that certain experiments lacked necessary controls, and the modelling part was inadequately described in the manuscript. Moreover, the experimental part of the manuscript is entirely based on single-molecule speckle tracking and some key results require confirmation by another experimental approach. Overall, an extensive amount of additional work would be required to address these points. Because of the policy of *eLife* to invite revisions only if they can be completed within 2-3 months, we cannot offer to consider this paper for further consideration.

Reviewers, however, provide several suggestions for how to improve the manuscript. Thus, if you can address these issues by performing additional experiments and by providing much better description of the models, we would be glad to consider a new submission on this topic for publication in *eLife*. In this case, the new submission would be evaluated by the three original reviewers.

Reviewer #1:

This manuscript reports a single-molecule speckle analysis of actin and selected other proteins in clathrin-mediated endocytosis in fission yeast. The authors provide evidence that these proteins display short residence times and rapid dynamics in endocytic patches. Analysis of single-molecule trajectories reveals that the motions of these proteins display differences that correlate with their functions in endocytosis.

The data presented in the manuscript appear of very good technical quality, and the study provides interesting insights into the molecular mechanisms of clathrin-mediated endocytosis. My main concern is that the data are somewhat redundant with the earlier FRAP analysis of endocytic components. For example, the studies by Kaksonen et al., (2003 and 2005) demonstrated (by using a Sla2del strain) that continuous actin polymerization occurs at the membrane of endocytosis structures in budding yeast with a comparable rate to the one determined in the present manuscript. This was also confirmed with another approach by Michelot et al., 2013. Thus, it is unclear for this reviewer, why the authors state several times in the manuscript that the 'common interpretation of the previous data is that the entire actin meshwork assembles and disassembles once during the total lifetime of the endocytic patch'. Moreover, the rapid turnover of clathrin was earlier demonstrated by FRAP analysis (Avionam et al., 2015). To merit publication in *eLife*, the authors should much better describe what it really novel in this study compared to earlier FRAP etc. experiments, and explain what fundamentally new information this work provides about the mechanisms of endocytosis. Otherwise, the manuscript is better suited for publication in a more specialized journal.

Moreover, the conclusions of the present study are entirely based on one method. Thus, the authors should use an alternative approach to confirm the main findings. They could e.g. carry out FRAP experiments on mEGFP-fusions of at least one actin filament binding protein and one nucleation promoting factor. If these proteins indeed display very short residence times in endocytic patches, this should be evident also form photobleaching experiments.

Reviewer #2:

In this manuscript, Lacy et al. aim to distinguish different mechanisms of molecule residence lifetimes using a combination of modeling and experiments. From the experimental standpoint, the authors provide quantitative measurements using a method they developed previously (Lacy et al., 2017) and I have no complaints with that. However, they use the model to identify possible mechanisms and here is where things get difficult.

1) What are they modeling? I read their Materials and methods, downloaded their MATLAB files, and tried to make sense of Figure 1. Given the sparse details in the Model simulations sections, I cannot make any sense of the model, the justification for the many assumptions and choice of parameters.

2) Then I downloaded the MATLAB file and tried to read it. At first glance, there is no documentation in the file, so I can't understand what's going on. After an inordinate amount of time spent on this. m file, I found the function simulator. It is a series of random and sorting functions! How do these functions represent the schematics in Figure 1A and B? Because the MATLAB file does not contain any representation of binding and unbinding events as mentioned in the Introduction, based on the information provided to me, the modeling component is not rigorous nor representative of the process they study.

3) The final issue is that the authors propose to distinguish different mechanisms from their data on the basis of this model. But since I don't understand the model, I cannot comment on the accuracy of the remainder of the work and therefore the scientific conclusions.

Reviewer #3:

Using single molecule labeling and imaging methods, Lacy et al. report that the lifetime of actin molecules and many of their regulators in actin patches is very short, on the order of seconds. This is much shorter than anticipated time based on the ~ 20 sec lifetime of actin patches. It is consistent with other studies in animal cells using similar methods that supported local turnover of dendritic actin structures. As yeast is a model organism for the study of actin cytoskeleton, this results will be noticed and important to the broad cytoskeleton field. I have however a few questions that I feel need to be addressed, mostly regarding the controls for accurate detection of such short-lived events.

1) Effect of photobleaching. In subsection “Single-molecule residence times of endocytic proteins are short” it is stated that photobleaching occurs over ~ 1 min so photobleaching is not important on the fast times scales of order 1 sec where actin turnover occurs. However this statement is not so obvious to me. In panel A of Figure 3—figure supplement 1 as well as Video 2 it's clear that photobleaching is significant over 5-10 sec, a time that could severely influence the calculated fraction of longer-lived spots (that would stay in the patch for a longer time and thus represent a bigger fraction of actin the patch). Further, given that in panel A the spots renew themselves by new monomers coming into the plane of focus, the photobleaching in the plane of focus could be even higher.

A control experiment on fixed cells is presented in panel C of Figure 3—figure supplement 1 where the spot lifetime is 15 sec on average (which is less than a min but longer than 1 sec). It is stated that the same imaging conditions were used as for live cells, 0.5 W/cm^2, but it's not clear of the frame acquisition rate and exposure were also kept the same. The binning of the histogram in panel C suggests that they were not and that the spot lifetime was about 10 exposures, corresponding to ~ 1 sec with the acquisition rate in live cells (panel B of Figure 3—figure supplement 1).

Can the authors address these apparent discrepancies?

2) Detection of spot lifetime in images. To address the difficulty of large number of spots in noisy images, the authors used a combination of automated and manual tracking of spot lifetime. However the spots are near the threshold of detection and in the provided videos and montages of Figure 1C and D it's hard to tell when the spot signal started and ended and whether blinking occurs. Providing examples of spot intensity versus time could help the readers evaluate the ability to detect and measure single molecule lifetimes, including addressing blinking effects and tracking errors. The effect of blinking could also be address nicely in fixed cells. The spot intensity should decrease rather abruptly in a depolymerization event and not slowly as might occur when the spot moves slowly out of focus.

3) How did the authors distinguish spots in patches at the cell middle that internalize away from the plane of focus versus spots in patches at the cell tip that could move parallel to the plane of focus? Does this effect influence the movement statistics in Figures 4-6?

[Editors’ note: what now follows is the decision letter after the authors submitted for further consideration.]

Thank you for submitting your article "Single-molecule turnover dynamics of actin and membrane coat proteins in clathrin-mediated endocytosis" for consideration by *eLife*. Your article has been reviewed by three peer reviewers, one of whom is a member of our Board of Reviewing Editors and the evaluation has been overseen by Vivek Malhotra as the Senior Editor. The reviewers have opted to remain anonymous.

The reviewers have discussed the reviews with one another and the Reviewing Editor has drafted this decision to help you prepare a revised submission.

All three reviewers found the manuscript significantly improved. They stated that the study provides important new information on the dynamics of actin and actin-associated proteins at the sites of clathrin-mediated endocytosis, and thus elucidates the mechanism of actin-based force generation in endocytosis. However, reviewers #2 and #3 raised few relatively minor points that should be addressed before publication.

Reviewer #2:

1) For completeness and making a more convincing case, I suggest that the authors provide one example of a video (single cells may be sufficient) for all tagged proteins and especially actin, for live and fixed cells as relevant.

2) Since, the authors work really at the limit of spot detection, providing examples of the filtered version of the video(s) that were used for measuring spot lifetime can also make the study more convincing. For non experts, it may be hard to see how lifetime measurement is feasible in the provided example of Acp1.

Reviewer #3:

1) In the Introduction, the authors state that, "As has been shown in other actin systems, continuous turnover of filaments allows a network to convert a larger amount of energy from ATP hydrolysis of actin polymerization into mechanical work over the meshwork's lifetime." And, " Based on these results, we suggest that the amount of force produced by the endocytic actin meshwork might be higher than has been previously estimated." These statements are not accompanied by any references. Given that this is a major finding, these statements should be supported by literature.

2) In light of point 1 above, some estimates of force production with and without actin turnover considerations would shed greater light on the impact of the work and make the manuscript more broadly accessible.

3) The modeling part of the work is now clearly explained, in that different models can give rise to the similar behavior (Figure 1). But now I'm wondering how does this contribute to the paper? The experiments and their discussions by themselves seem more solid and it seems that the modeling aspect really doesn't identify hypotheses or bring significant value to the interpretation of the data. It is true that's multiple mathematical mechanisms can give rise to similar looking functions but unless one is able to distinguish one mechanism from the other with a large degree of certainty, it's best to leave this portion out.

---

## [Author Response]

[Editors’ note: the author responses to the first round of peer review follow.]

Reviewer #1:

[…] The data presented in the manuscript appear of very good technical quality, and the study provides interesting insights into the molecular mechanisms of clathrin-mediated endocytosis. My main concern is that the data are somewhat redundant with the earlier FRAP analysis of endocytic components. For example, the studies by Kaksonen et al., (2003 and 2005) demonstrated (by using a Sla2del strain) that continuous actin polymerization occurs at the membrane of endocytosis structures in budding yeast with a comparable rate to the one determined in the present manuscript. This was also confirmed with another approach by Michelot et al., 2013. Thus, it is unclear for this reviewer, why the authors state several times in the manuscript that the 'common interpretation of the previous data is that the entire actin meshwork assembles and disassembles once during the total lifetime of the endocytic patch'. Moreover, the rapid turnover of clathrin was earlier demonstrated by FRAP analysis (Avionam et al., 2015). To merit publication in eLife, the authors should much better describe what it really novel in this study compared to earlier FRAP etc. experiments, and explain what fundamentally new information this work provides about the mechanisms of endocytosis. Otherwise, the manuscript is better suited for publication in a more specialized journal.

We thank the reviewer for their positive comments on the technical and experimental quality.

We emphasize that while our results are consistent with those FRAP experiments, single-molecule observations extend beyond some important limitations of FRAP data. Indeed interpretation of FRAP data can give access to the time-constant for recovery, which is a combination of the apparent on- and off-rate constants, but requires assumptions (typically bi-molecular binding with mass-action kinetics) that are sometimes difficult to verify. Our single-molecule approach gives a distribution of residence times instead of one summary number, and is independent of any assumption or modeling. For example, our data clearly show that the dynamics of the endocytic proteins we have measured does not follow simple mass action kinetics (otherwise, the residence time distributions would be exponential). Distributions also allow a more detailed view of complex behaviors, as in the case of Fim1 where we observed a heterogeneous distribution with a sub-population of very short residence time events (Figure 3).

In the case of CME in wild-type cells, because the number of molecules is continually and rapidly changing, interpretation of FRAP data is difficult and does not have high precision. To make it easier, previous FRAP studies have used mutants or in vitro system to observe longer-lasting comet tails (e.g. Kaksonen et al., 2003 and Michelot et al., 2013). The numbers they measured are informative, but might not be the same as in a wild-type context.

We believe that most people in the field do not include turnover in their thinking about endocytosis. Most papers that have been recently published, including reviews and mathematical models from the labs who published the FRAP papers cited by Reviewer 1, have assumed that endocytic proteins assemble once while the endocytic pit is elongated, then disassemble after the vesicle is pinched off. For example, this no-turnover hypothesis is particularly clear in the computational simulations presented in Mund et al. (Cell, 2018) and Akamatsu et al. (BioRxiv, 2019). In these models (and other papers), it is believed that the force exerted on the barbed ends of actin filaments slows down their polymerization, which makes turnover incompatible with the forces exerted and the timescale of endocytosis.

We have revised the Introduction and Discussion sections to better explain these limitations of previous studies and highlight the novel insights of our approach.

Moreover, the conclusions of the present study are entirely based on one method. Thus, the authors should use an alternative approach to confirm the main findings. They could e.g. carry out FRAP experiments on mEGFP-fusions of at least one actin filament binding protein and one nucleation promoting factor. If these proteins indeed display very short residence times in endocytic patches, this should be evident also form photobleaching experiments.

The reviewer agreed above that our data are consistent with previous FRAP experiments with actin and clathrin. We have now added new FRAP experiments on other proteins (Wsp1p, Myo1p, Fim1p, Acp1p) which have not been previously reported. These experiments are presented in Figure 2F and Figure 2—figure supplement 2.

Similar to previous reports on other endocytic components, we observe rapid recovery of fluorescent signal after photobleaching of the endocytic patch. Importantly, the recovery time is on the order of 1-2 sec and the bleached patches reach a maximum intensity similar to unbleached patches, indicating that the recruited molecules are rapidly turned over and the whole assembly is turned over and renewed multiple times during the course of a CME event. This observation is consistent with our single-molecule data and strengthens our conclusions from that technique.

This data is presented in Figure 2F and Figure 2—figure supplement 2, and see revised text in Results, Discussion, and Materials and methods.

Reviewer #2:

In this manuscript, Lacy et al. aim to distinguish different mechanisms of molecule residence lifetimes using a combination of modeling and experiments. From the experimental standpoint, the authors provide quantitative measurements using a method they developed previously (Lacy et al., 2017) and I have no complaints with that. However, they use the model to identify possible mechanisms and here is where things get difficult.

We thank the reviewer for their positive comments about the experimental data. However, it seems that our discussion and presentation of the simulation components caused significant confusion. We apologize for that and have made revisions as detailed below.

1) What are they modeling? I read their Materials and methods, downloaded their MATLAB files, and tried to make sense of Figure 1. Given the sparse details in the Model simulations sections, I cannot make any sense of the model, the justification for the many assumptions and choice of parameters.

As noted above, we apologize for the confusion. We did not aim to model the full complexity of molecular binding/unbinding and actin dynamics in CME but only intended to illustrate the basic logic of “turnover” and “residence times”. These calculations simply show that a similar time profile of accumulation of arbitrary objects can be achieved with very different logical rules, regardless of the underlying molecular processes.

We have added better documentation in the Matlab script (Source code 1) and revised the descriptions in Results and Materials and methods.

2) Then I downloaded the MATLAB file and tried to read it. At first glance, there is no documentation in the file, so I can't understand what's going on. After an inordinate amount of time spent on this. m file, I found the function simulator. It is a series of random and sorting functions! How do these functions represent the schematics in Figure 1A and B? Because the MATLAB file does not contain any representation of binding and unbinding events as mentioned in the Introduction, based on the information provided to me, the modeling component is not rigorous nor representative of the process they study.

We have revised the text to better explain this component and improved the documentation of the code. As noted above, we clarify that it is an exercise in mathematical logic rather than any attempt of modeling of the molecular mechanisms. For example, the text now refers to “arrival and departure times” of “objects” or “components” rather than “binding and unbinding” of “molecules” or “monomers”.

We hope the editor and reviewer will agree that these calculations provide a useful illustrative exercise, even without explicit molecular details. A rigorous and complete molecular simulation of CME is beyond the scope of this paper, although it is a focus of ongoing work in our lab.

We have added better documentation in the Matlab script (Source code 1) and revised the descriptions in Results and Materials and methods.

3) The final issue is that the authors propose to distinguish different mechanisms from their data on the basis of this model. But since I don't understand the model, I cannot comment on the accuracy of the remainder of the work and therefore the scientific conclusions.

Again, we apologize for the confusion but we emphasize that the scientific conclusions of our work do not rely on the modeling presented at the start of the Results. The modeling was used to illustrate to readers that the previously-observed bulk assembly and disassembly time profiles can be generated by very different underlying behaviors. Our single-molecule speckle microscopy results directly support a model of continuous turnover and disprove a model of single-turnover assembly, without any explicit need for simulations to aid the interpretation.

As in the original manuscript, this modeling exercise is only mentioned briefly in the Discussion and is not necessary to reach our conclusions, but we have now revised the text as noted in Results.

Reviewer #3:

Using single molecule labeling and imaging methods, Lacy et al. report that the lifetime of actin molecules and many of their regulators in actin patches is very short, on the order of seconds. This is much shorter than anticipated time based on the ~ 20 sec lifetime of actin patches. It is consistent with other studies in animal cells using similar methods that supported local turnover of dendritic actin structures. As yeast is a model organism for the study of actin cytoskeleton, this results will be noticed and important to the broad cytoskeleton field. I have however a few questions that I feel need to be addressed, mostly regarding the controls for accurate detection of such short-lived events.1) Effect of photobleaching. In subsection “Single-molecule residence times of endocytic proteins are short” it is stated that photobleaching occurs over ~ 1 min so photobleaching is not important on the fast times scales of order 1 sec where actin turnover occurs. However this statement is not so obvious to me. In panel A of Figure 3—figure supplement 1 as well as Video 2 it's clear that photobleaching is significant over 5-10 sec, a time that could severely influence the calculated fraction of longer-lived spots (that would stay in the patch for a longer time and thus represent a bigger fraction of actin the patch). Further, given that in panel A the spots renew themselves by new monomers coming into the plane of focus, the photobleaching in the plane of focus could be even higher.

We thank the reviewer for this comment, as the concern about photobleaching is an important consideration. Our characterization of bleaching in the original text may have been overstated and we did not explain our interpretation sufficiently clearly.

Because the illumination field is partial-TIRF (also known as HILO – Highly inclined and laminated optical sheet – or VAEM – variable-angle epifluorescence microscopy) and not complete TIRF, we do not believe there is a big difference between bleaching “in the plane of focus” and in the rest of the cell because a large fraction of the cell volume is illuminated. In fact, some images were focused above the cell base and we observed events near the cell tips, indicating that half the cell or more is illuminated. Although spots appear when monomers enter the plane of focus, they are already in the illumination volume, which is why the overall number of molecules decays over time instead of a constant renewal of unbleached fluorophores.

The decay of spots per frame in Figure 3—figure supplement 1A does change in proportion to the difference of laser power but the residence time distributions in panel B are indistinguishable. This would not be observed if the residence times were significantly affected by photobleaching rates.

We have revised the text in Results to better address these points.

A control experiment on fixed cells is presented in panel C of Figure 3—figure supplement 1 where the spot lifetime is 15 sec on average (which is less than a min but longer than 1 sec). It is stated that the same imaging conditions were used as for live cells, 0.5 W/cm^2, but it's not clear of the frame acquisition rate and exposure were also kept the same. The binning of the histogram in panel C suggests that they were not and that the spot lifetime was about 10 exposures, corresponding to ~ 1 sec with the acquisition rate in live cells (panel B of Figure 3—figure supplement 1).Can the authors address these apparent discrepancies?

We apologize that the original description was not clear. All the imaging conditions and camera settings, including frame acquisition and exposure time, were indeed the same. The reviewer’s suggestion that these lifetimes correspond to ~1 sec is incorrect. The difference in binning size is simply for the purpose of displaying the data, as the number of spots tracked in the fixed-cell images was much smaller. The time axes and data are displayed properly and there is no discrepancy; the average lifetime was indeed 15 sec with several events tracked over 30 to 60 sec.

We have revised the text to clarify these points in Materials and methods and Results and Supplemental figure legend.

2) Detection of spot lifetime in images. To address the difficulty of large number of spots in noisy images, the authors used a combination of automated and manual tracking of spot lifetime. However the spots are near the threshold of detection and in the provided videos and montages of Figure 1C and D it's hard to tell when the spot signal started and ended and whether blinking occurs. Providing examples of spot intensity versus time could help the readers evaluate the ability to detect and measure single molecule lifetimes, including addressing blinking effects and tracking errors. The effect of blinking could also be address nicely in fixed cells. The spot intensity should decrease rather abruptly in a depolymerization event and not slowly as might occur when the spot moves slowly out of focus.

We thank the reviewer for raising this comment, as this point was not explained clearly enough in the original submission. However, we clarify that we did not use “a combination of automated and manual tracking of spot lifetime” for our single-molecule experiments in live cells, as all the spot detection and tracking for single-molecule data was performed in a fully automated manner. We only used semi-manual tracking for fixed cells (as in Figure 3—figure supplement 1C) and on GFP patches (as in Figure 2C). Semi-manual tracking was used in GFP-labeled cells because there are too many overlapping patches to reliably track isolated patches automatically; this is a standard method we and others in the field have used.

We provided several indirect measures of the tracking algorithm’s quality and robustness (see Figure 3—figure supplement 3 and Materials and methods “Characterization of tracking parameters and data quality”). Unfortunately, this question could not be easily addressed in fixed cells due to increased background fluorescence in these samples, which required different spot detection and semi-manual tracking (making any comparison of technical accuracy invalid).

Due to the tracking algorithm workflow, the intensity of spots at time points which are not detected or not included in a track (whether due to blinking, insufficient signal/noise to meet thresholds, etc.) are not calculated and are not retained in the output data so we cannot directly display the intensity over time in the way the reviewer suggests. We felt it would be misleading to present readers with intensity profiles calculated by a different method which could calculate the intensity at “missed” time points as evidence of the tracking algorithm’s accuracy. But, due to the reviewer’s concern we now provide this as a new part of Figure 3—figure supplement 1.

We also refer to Figure 3—figure supplement 3, where we showed the effects of varying the tracking parameters and the artifacts introduced by over- or under-linking (i.e. improperly accounting for blinking and signal/noise effects). Gaps in a track are due to photophysical blinking of the dye or to missed detections of the tracking algorithm due to the low signal/noise ratio and must be linked with appropriate sensitivity. The parameter values were chosen to maximize the track lifetimes and minimize these artifacts.

We also note that increasing the laser power (which not only increases the signal/noise ratio but also increases rates for photophysical blinking/bleaching) had no effect on track lifetimes when using the same tracking parameters (Figure 3—figure supplement 1).

We have revised the text to better discuss these issues (subsection “Single-molecule residence times of endocytic proteins are short”).

3) How did the authors distinguish spots in patches at the cell middle that internalize away from the plane of focus versus spots in patches at the cell tip that could move parallel to the plane of focus? Does this effect influence the movement statistics in Figures 4-6?

We did not distinguish spots at the cell middle versus spots at the cell tip, and have not attempted any alignment or averaging of trajectories.

As explained in the text, we attribute large scale motions (>500 nm) to molecules associated with the diffusing vesicle; after scission the vesicle moves randomly in three dimensions regardless of its origin or the relative orientation of the membrane. Given the limits of our resolution we do not rely heavily on small differences in short motions (<100 nm), which reflect a mix of 2D-diffusive membrane-bound molecules or molecules moving with the invaginating CCP, in various orientations. Interpretation of these features would be complicated by the localization precision of our algorithms (~50 nm).

However, because all samples were imaged in the same partial-TIRF mode, this factor has a similar effect on all the datasets and cannot explain the differences in behaviors between proteins.

We have added a statement acknowledging this limitation.

[Editors' note: the author responses to the re-review follow.]

All three reviewers found the manuscript significantly improved. They stated that the study provides important new information on the dynamics of actin and actin-associated proteins at the sites of clathrin-mediated endocytosis, and thus elucidates the mechanism of actin-based force generation in endocytosis. However, reviewers #2 and #3 raised few relatively minor points that should be addressed before publication.

We thank the reviewers and the editors for their positive comments. We have edited the manuscript and provided additional Supplementary files as indicated in our responses below.

In addition to these comments, we have also added the Key Resources Table in the Materials and methods. Because the Key Resources Table includes information on yeast strains, we have removed Table 4 and incorporated that information here instead.

Reviewer #2:

1) For completeness and making a more convincing case, I suggest that the authors provide one example of a video (single cells may be sufficient) for all tagged proteins and especially actin, for live and fixed cells as relevant.

We have included representative videos for all samples as suggested.

2) Since, the authors work really at the limit of spot detection, providing examples of the filtered version of the video(s) that were used for measuring spot lifetime can also make the study more convincing. For non experts, it may be hard to see how lifetime measurement is feasible in the provided example of Acp1.

We did not apply any additional filtering or processing to the videos before analysis. The videos provided here are inverted contrast and compressed in AVI format for web submission, but the raw TIFF files are available upon request. We have added a note to explain this in the Supplementary file legends.

Reviewer #3:

1) In the introduction, the authors state that, "As has been shown in other actin systems, continuous turnover of filaments allows a network to convert a larger amount of energy from ATP hydrolysis of actin polymerization into mechanical work over the meshwork's lifetime." And, " Based on these results, we suggest that the amount of force produced by the endocytic actin meshwork might be higher than has been previously estimated." These statements are not accompanied by any references. Given that this is a major finding, these statements should be supported by literature.

We have edited these sentences and added references. We clarify that these are two different, though related, points. Previous studies have found that actin filament turnover is necessary for sustained force production and enables greater force over time than a meshwork generated by a single assembly phase. We also emphasize the conclusion that a higher number of molecules consumed implies more total available energy through ATP hydrolysis and other biochemical mechanisms, however we do not make any calculations of actual forces and energy.

See the Introduction, and point 2 below.

2) In light of point 1 above, some estimates of force production with and without actin turnover considerations would shed greater light on the impact of the work and make the manuscript more broadly accessible.

Because the force of actin polymerization depends on a variety of factors (see our recent review, Lacy 2018 FEBS Letters), we did not attempt to calculate force outputs of the actin meshwork and other CME components. However, we infer that if the continuously turned over CME meshwork consumes up to five times more molecules of ATP-actin, myosin I, and other components than previously estimated, then the total amount of energy available to be expended as physical work would also up to five times higher than previous estimates. This is not really an estimate of the actual force output, but is suggested as a greater upper limit of available energy for force generation than might be expected based only on the total number of molecules present. The endocytic patch contains approximately 6,500 actin monomers at its peak (Sirotkin, 2010 MBoC, Berro, 2010 MBoC), but based on the residence times we report here, a single patch might consume more than 30,000 actin subunits over time. Further modeling and experimental work is needed to understand how the available chemical energy is converted into mechanical work.

We note that recent theoretical work based on the dendritic nucleation model (which includes filament turnover) suggests the actin meshwork is capable of producing sufficient forces for CME, ~2500 pN (Nickaeen et al., 2019).

We also point out that a meshwork that is rapidly and continuously turned over may have additional important mechanisms allowing generation and distribution of forces that are not considered by previous models, which makes a direct comparison of force outputs based on simple calculations more difficult and perhaps less meaningful.

We have made some edits to the text which we believe addresses both point 1 and point 2, see Introduction, and subsection “Implications of actin turnover for force production models”.

3) The modeling part of the work is now clearly explained, in that different models can give rise to the similar behavior (Figure 1). But now I'm wondering how does this contribute to the paper? The experiments and their discussions by themselves seem more solid and it seems that the modeling aspect really doesn't identify hypotheses or bring significant value to the interpretation of the data. It is true that's multiple mathematical mechanisms can give rise to similar looking functions but unless one is able to distinguish one mechanism from the other with a large degree of certainty, it's best to leave this portion out.

The goal of the modeling presented in this paper is to demonstrate that the commonly believed idea, namely the discrete assembly/disassembly model illustrated where the actin assembles while the clathrin-coated pit elongates then disassembles after the vesicle is pinched off, is not compatible with our single-molecule residence time data. We think that hypothesis rejection is a very powerful use of mathematical modeling, and arguably the only rigorous conclusion one can draw from a model or an experiment (Berro, 2018; Popper, 1959). For this reason, we strongly believe our modeling efforts belong in this paper. In addition, we propose a mechanism that is compatible with our single-molecule data and discuss its implications, even though we acknowledge that this model is not the only model that could explain the data, and that it does not explicitly detail the molecular mechanisms that drive the short residence of the single molecules we measured. This model of a dendritic network with rapid and continuous turnover should serve as an inspiration for the community to design future experiments and models that will aim to uncover the underlying molecular mechanisms.

**References**

Berro J. 2018. “Essentially, all models are wrong, but some are useful”— a cross-disciplinary agenda for building useful models in cell biology and biophysics. Biophys Rev 10:1637–1647. doi:10.1007/s12551-018-0478-4

Popper KR. 1959. The logic of scientific discovery. Hutchinson.